# Late Paleocene $CO_2$ drawdown, climatic cooling and terrestrial denudation in the southwest Pacific

**Christopher J. Hollis**[1,2], **Sebastian Naeher**[1], **Christopher D. Clowes**[1], **B. David A. Naafs**[3], **Richard D. Pancost**[3], **Kyle W. R. Taylor**[3], **Jenny Dahl**[1], **Xun Li**[1], **G. Todd Ventura**[1,4], **and Richard Sykes**[1]

[1]GNS Science, Lower Hutt, 5040, New Zealand
[2]School of Geography, Environment and Earth Sciences, Victoria University of Wellington, Wellington, New Zealand
[3]Organic Geochemistry Unit, School of Chemistry, School of Earth Sciences and Cabot Institute for the Environment, University of Bristol, Bristol, UK
[4]Department of Geology, Saint Mary's University, Halifax, Nova Scotia, Canada

**Correspondence:** Christopher J. Hollis (chris.hollis@vuw.ac.nz) and Sebastian Naeher (s.naeher@gns.cri.nz)

**Abstract.** TS1 Late Paleocene deposition of an organic-rich sedimentary facies on the continental shelf and slope of New Zealand and eastern Australia has been linked to short-lived climatic cooling and terrestrial denudation following sea level fall. Recent studies confirm that the organic matter in this facies, termed "Waipawa organofacies", is primarily of terrestrial origin, with a minor marine component. It is also unusually enriched in $\delta^{13}C$ CE1. In this study we address the cause of this enrichment. For Waipawa organofacies and its bounding facies in the Taylor White section, Hawke's Bay, paired palynofacies and carbon isotope analysis of heavy liquid-separated density fractions indicate that the heaviest $\delta^{13}C$ values are associated with degraded phytoclasts (woody plant matter) and that the $^{13}C$ enrichment may be partly due to lignin degradation. Compound-specific stable carbon isotope analyses of samples from the Taylor White and mid-Waipara (Canterbury) sections display similar trends and further reveal a residual $^{13}C$ enrichment of $\sim 3‰$ TS2 in higher plant biomarkers (long chain $n$-alkanes and fatty acids) and a $\sim 2‰–5‰$ TS3 change in subordinate marine biomarkers. Using the relationship between atmospheric $CO_2$ and $C_3$ plant tissue $\delta^{13}C$ values, we determine that the 3‰ increase in terrestrial $\delta^{13}C$ may represent a $\sim 40\%$ decrease in atmospheric $CO_2$.

Refined age control for Waipawa organofacies indicates that deposition occurred between 59.2 and 58.5 Ma, which coincides with an interval of carbonate dissolution in the deep sea that is associated with a Paleocene oxygen isotope maximum (POIM, 59.7–58.1 Ma) and the onset of the Paleocene carbon isotope maximum (PCIM, 59.3–57.4 Ma). This association suggests that Waipawa deposition occurred during a time of cool climatic conditions and increased carbon burial. This relationship is further supported by published $TEX_{86}$-based sea surface temperatures that indicate a pronounced regional cooling during deposition. We suggest that reduced greenhouse gas emissions from volcanism and accelerated carbon burial, due to tectonic factors, resulted in short-lived global cooling, growth of ephemeral ice sheets and a global fall in sea level. Accompanying erosion and carbonate dissolution in deep-sea sediment archives may have hidden the evidence of this "hypothermal" event until now.

## 1 Introduction

The Paleocene epoch (66–56 Ma) is bookended by the two most extreme biotic and climatic events of the Cenozoic (Fig. 1): the Cretaceous–Paleogene (K–Pg) mass extinction and the Paleocene–Eocene Thermal Maximum (PETM; Zachos et al., 2008). The intervening 10 Myr are a complex series of climate oscillations that culminated in the warmest temperatures of the Cenozoic (Zachos et al., 2008; Komar et al., 2013; Taylor et al., 2018; Hollis et al., 2019). Paleocene climate variability has been linked to various factors affecting $CO_2$ levels, including events relating directly to the K–Pg bolide impact and mass extinction (Schulte et al., 2010),

CO$_2$ emissions from volcanism (Westerhold et al., 2011), the exhumation (Beck et al., 1995) and burial (Kurtz et al., 2003) of organic carbon due to tectonic processes, and biological productivity (Corfield and Cartlidge, 1992). A peak in benthic foraminiferal $\delta^{13}$C values at 59–57 Ma, referred to as the Paleocene carbon isotope maximum (PCIM), is thought to represent a period of enhanced carbon burial, perturbing the global carbon cycle (Fig. 1a). The process may have been driven by North American uplift, which led to large epeiric seas being transformed into extensive carbon-sequestering peat deposits (Kurtz et al., 2003). Other studies suggest that changes in ocean circulation caused an increase in marine productivity and oceanic carbon burial, either as a positive feedback to a long-term cooling trend (Corfield and Cartlidge, 1992; Corfield and Norris, 1996) or due to the opening of pathways for deep-water circulation (Batenburg et al., 2018). Late Paleocene climatic cooling may have also promoted carbon burial in the form of biogenic methane (hydrates) on continental margins (Dickens, 2003) or as high-latitude permafrost (DeConto et al., 2012).

Despite limited and conflicting data for the Paleocene, CO$_2$ levels in the early Paleogene approach those that we may expect in coming centuries (Foster et al., 2017). Therefore, the existence of an episode in which extensive carbon burial is linked to climatic cooling warrants investigation as a guide to processes that might cool the planet by reducing the flux of CO$_2$ into the atmosphere. However, the temperature response to the PCIM is poorly understood and in deep-sea records there appears to be an offset between the two, with the most positive benthic foraminiferal $\delta^{18}$O values being at the onset ($\sim$ 59 Ma) rather than the peak ($\sim$ 58 Ma) of the PCIM (Fig. 1b). This apparent offset may be related to pervasive deep-sea carbonate dissolution across this 1 Myr interval (Westerhold et al., 2011; Littler et al., 2014), which has the potential to distort the benthic foraminiferal stable oxygen isotope ($\delta^{18}$O) records (Fig. 1c).

In contrast to these deep-sea $\delta^{18}$O records, studies of continental margin sediments in the southwest Pacific reveal clearer evidence for pronounced cooling in the late Paleocene (Bijl et al., 2009; Contreras et al., 2014; Hollis et al., 2012, 2014) (Fig. 1d), which has been linked to a fall in sea level and widespread deposition of organic-rich marine sediments (Schiøler et al., 2010; Hollis et al., 2014). Recent integrated palynofacies and geochemical studies of the distinctive organic-matter (OM) assemblage in these sediments, termed "Waipawa organofacies" by Hollis et al. (2014), have found that it is primarily of terrestrial origin, comprising mainly degraded wood fragments or phytoclasts (Field et al., 2018; Naeher et al., 2019). In addition, deposition occurred rapidly with a compacted mass accumulation rate up to 10 times greater than the background rate (Hollis et al., 2014; Hines et al., 2019; Naeher et al., 2019). In this study, we focus on another primary feature of Waipawa organofacies: the bulk OM is highly enriched in $^{13}$C, with a mean $\delta^{13}$C value of [TS4]$-20$‰, which is $\sim$ 7‰ heavier than OM in sediments directly above and below (Schiøler et al., 2010; Hollis et al., 2014; Naeher et al., 2019).

In the absence of evidence for major changes in terrestrial vegetation (Contreras et al., 2014), the lack of isotopically heavy C$_4$ plants that only evolved in the Neogene (Urban et al., 2010) and no obvious changes in aridity or precipitation (Lomax et al., 2019; Schlanser et al., 2020), we explore the possibility that this $^{13}$C enrichment of bulk OM reflects a short-lived drawdown in atmospheric CO$_2$, reflecting the relationship in carbon isotope discrimination between atmospheric CO$_2$ and C$_3$ plant biomass (Schubert and Jahren, 2012, 2018; Cui and Schubert, 2016, 2017, 2018). For this purpose, we analysed the $\delta^{13}$C values of specific organic fractions (palynodebris) and selected biomarkers from Waipawa organofacies and the "background" bounding facies at two sites (Taylor White and mid-Waipara sections) to identify the source of $^{13}$C enrichment. In addition, we evaluate the roles that lignin degradation (e.g. van Bergen and Poole, 2002) and OM sulfurization (e.g. Sinninghe Damsté et al., 1998; Rosenberg et al., 2018) may have played. From these analyses, we estimate the magnitudes of the $\delta^{13}$C excursion in both primary terrestrial and marine OM and use these values to infer changes in the concentration of atmospheric CO$_2$.

## 2  Sites and sections studied

This study of Waipawa organofacies includes a compilation of data from 10 onshore sections and 1 onshore and 6 offshore drill holes from New Zealand and the southwest Pacific (Fig. 2). The sections and site locations are described in Sect. S1 in the Supplement (Fig. S1, Table S1). Waipawa organofacies is most readily identified by $\delta^{13}C_{OM}$ values higher than $-24.5$‰ (Hollis et al., 2014). Enrichment in total organic carbon (TOC) is also a useful guide (Fig. 3), although there is wide variation between sections due to depositional setting (Hollis et al., 2014). Waipawa organofacies is a defining feature of the Waipawa Formation in the East Coast and Northland basins (Moore, 1988; Isaac et al., 1994; Field and Uruski, 1997; Hollis et al., 2006) and the Tartan Formation in the Canterbury and Great South basins (Cook et al., 1999; Schiøler et al., 2010). Equivalent facies have also been identified in the Taranaki and West Coast basins (Killops et al., 2000) and the southwest Tasman Sea (Röhl et al., 2004; Hollis et al., 2014). The background facies in most of the East Coast Basin comprise the underlying Whangai Formation, an organic-poor siliceous to slightly calcareous mudstone, and the overlying Wanstead Formation, an organic-poor non-calcareous to moderately calcareous mudstone (Moore, 1988; Field and Uruski, 1997). In the Marlborough Sub-basin, these two units are replaced by more pelagic facies: siliceous micrites of the Mead Hill Formation and the basal Amuri Limestone (Field and Uruski, 1997; Hollis et al., 2005). In the Great South Basin, the

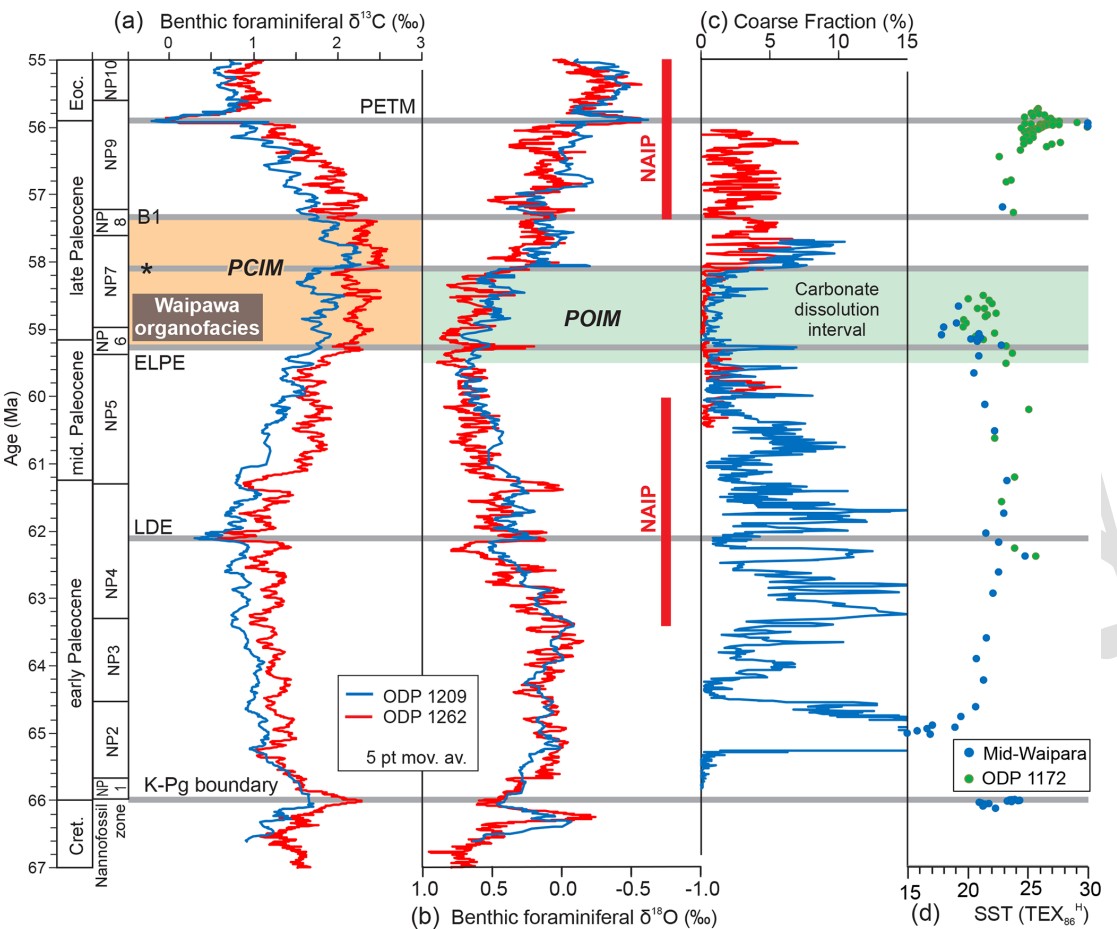

**Figure 1.** Paleocene variation in benthic foraminiferal carbon **(a)** and oxygen **(b)** stable isotopes and sediment coarse-fraction ($> 63\,\mu m$) percentage **(c)** for ODP sites 1209 and 1262 (from Barnet et al., 2019) compared with **(d)** variation in $TEX_{86}^{H}$-derived SST estimates from ODP Site 1172 and the mid-Waipara River section (from Hollis et al., 2012, 2014, 2019). Climatic and biotic events highlighted are the Cretaceous–Paleogene (K–Pg) boundary, Late Danian Event (LDE), early late Paleocene event (ELPE), Paleocene oxygen isotope maximum (POIM), Paleocene carbon isotope maximum (PCIM) and Paleocene–Eocene Thermal Maximum (PETM). Timing of the North Atlantic Igneous Province (NAIP) eruptive phases and Waipawa organofacies deposition is also shown.

bounding formations are the underlying Wickliffe and overlying Laing formations, both of which are more siliciclastic than their East Coast Basin counterparts (Cook et al., 1999).

## 3 Palynofacies and geochemical analyses

We undertook palynofacies and geochemical analyses of rock samples from the Waipawa organofacies and bounding facies in the following stratigraphic sections in northern and eastern New Zealand: Black's Quarry, Taylor White, Glendhu Rocks (Pahaoa River mouth), Chancet Rocks, Ben More Stream, Mead Stream and mid-Waipara River (Table S2). We combine these new results with published data from the following sections and drill holes: Te Hoe River (Schiøler et al., 2010); Tawanui, Angora Road and mid-Waipara River (Taylor, 2011; Hollis et al., 2014); Taylor White (Naeher et al., 2019); Orui-1A onshore drill hole

(Field et al., 2018); Mead Stream (Hollis et al., 2005); Galleon-1 (Schiøler, 2011), Toroa-1, Pakaha-1, Kawau-1A and Hoiho-1C offshore drill holes (Raine et al., 1993; Schiøler et al., 2010); and ODP Site 1172, East Tasman Plateau (Hollis et al., 2014; Bijl et al., 2021). We also draw on published stable carbon isotope data from spot samples of the Waipawa Formation and bounding formations at Te Weraroa Stream, Angora Stream and Te Puia as well as from Paleocene coaly rock samples (Sykes and Zink, 2012; Sykes et al., 2012).

Homogenized, representative sample aliquots were prepared and palynological and geochemical analyses undertaken using the methods described by Naeher et al. (2019). These included procedures for palynological analysis to determine palynofacies composition (Tables S2 and S3); bulk pyrolysis to quantify OM richness using a source rock analyser (SRA) (Table S2); bulk carbon content and stable iso-

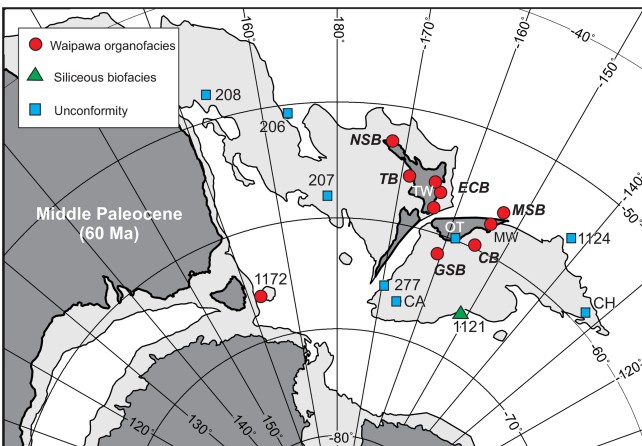

**Figure 2.** Localities where Waipawa organofacies is present or represented by an unconformity or a correlated siliceous biofacies on a middle-Paleocene paleogeographic reconstruction (from Hollis et al., 2014). Locality abbreviations: NSB, North Slope Basin; TB, Taranaki Basin; TW, Taylor White section; ECB, East Coast Basin; MSB, Marlborough Sub-basin; MW, mid-Waipara section; CB, Canterbury Basin; OT, onshore Otago; GSB, Great South Basin; CH, Chatham Island; CA, Campbell Island. Numbers refer to DSDP or ODP sites. See Fig. S1 for present-day section locations.

tope analysis of decalcified rock powders using elemental analysis–isotope ratio mass spectrometry (EA–IRMS) (Table S2); analysis of kerogen phenol and thiophene concentrations by pyrolysis–gas chromatography–mass spectrometry (Py–GC–MS) (Table S4); and solvent extraction and analysis of compound-specific stable carbon isotopes of lipid biomarkers by gas chromatography–combustion–isotope ratio mass spectrometry (GC–C–IRMS) (Table S5). Additional procedures employed in this study are described in the following sections.

## 3.1 Density fractionation

Total organic residues obtained from palynological preparations of eight Waipawa and two CE2 Whangai samples from the Taylor White section were solvent extracted to remove any bitumen present and then processed by density fractionation with the aim of identifying the palynofacies source or sources for $^{13}$C enrichment in Waipawa organofacies (Table S3). The preparations (3–597 mg) were sieved into grain-size fractions < 6 and ≥ 6 μm. Both grain-size fractions were processed by density separation using sodium polytungstate (Na$_6$[H$_2$W$_{12}$O$_{40}$]; high-purity SPT-0, TC-Tungsten Compounds GmbH, Germany) in deionized water. Five to eight density fractions per grain-size fraction were obtained, from < 1.2 to > 1.5 g cm$^{-3}$. Palynofacies analysis was carried out on 86 density fractions in the ≥ 6 μm suite. Carbon isotope analysis was carried out on 77 samples in this suite and 10 samples in the < 6 μm suite (Table S3).

## 3.2 Solvent extraction, biomarker and carbon stable isotope analyses

To investigate the source and composition of OM and to help reconstruct depositional and environmental conditions, lipid biomarkers and the carbon isotope composition of the total saturated and aromatic hydrocarbon fractions in solvent extracts (bitumen) from the Taylor White section were previously analysed at Applied Petroleum Technology (APT) in Oslo, Norway, with the methods and data reported by Naeher et al. (2019). Only the carbon isotope values of the total saturated and aromatic fractions (Table S4) and representative chromatograms are presented in this paper.

Compound-specific carbon isotope analyses of selected isoprenoids (pristane and phytane), $n$-alkanes ($n$C$_{18}$–$n$C$_{33}$) (Table S4) and fatty acids (Table S6) were undertaken in the Organic Geochemistry Laboratory at GNS Science and the Organic Geochemistry Unit (OGU) at the University of Bristol, UK. Samples from the Taylor White section were prepared using the analytical procedures reported in Naeher et al. (2012, 2014) with some modifications. In brief, 7–40 g powdered rock was extracted (four times) with dichloromethane (DCM) / methanol (MeOH) (3 : 1, $v/v$) by ultrasonication for 20 min each time. Elemental sulfur was removed by activated copper. The total lipid extracts (TLEs) were divided into saturated and aromatic hydrocarbon and polar compound fractions via liquid chromatography over silica columns using $n$-hexane, $n$-hexane / DCM (7 : 3, $v/v$), and DCM / MeOH (1 : 1, $v/v$), respectively. An aliquot of the polar fraction was derivatized with BSTFA ($N$,$O$-bis(trimethylsilyl)trifluoroacetamide) (Sigma Aldrich) for 1 h at 80 °C prior to analysis.

For compound-specific carbon isotope analyses of pristane (Pr), phytane (Ph) and $n$-alkanes (Table S4) we undertook molecular sieving (Dawson et al., 2005; Grice et al., 2008; Aboglila et al., 2010) of free and desulfurized saturated fractions of samples from the Taylor White section. Saturated hydrocarbon fractions dissolved in cyclohexane were added to an activated 5 Å molecular sieve (Alltech) and heated for 8 h at 80 °C. Branched and cyclic compounds were recovered by extraction (five times) with cyclohexane. $n$-Alkanes were recovered by dissolution of the sieve with 30 % HF, followed by neutralization with a saturated NaHCO$_3$ solution. The resulting fractions were analysed using an Isoprime 100 GC–C-IRMS system at the University of Bristol. Injection volume was 1 μL onto to a Zebron-I nonpolar column (50 m × 0.32 mm inner diameter, 0.10 μm film thickness). The GC oven programme was 3 min hold at 70 °C, heating to 130 °C at 20 °C min$^{-1}$ and then to 300 °C at 4 °C min$^{-1}$, and a final hold at 300 °C for 25 min. Samples were measured in duplicate and $\delta^{13}$C values converted to Vienna Pee Dee Belemnite (VPDB) by bracketing with CO$_2$ of known $\delta^{13}$C value. Instrument stability was monitored by regular analysis of an in-house fatty-acid CE3 methyl ester standard mixture.

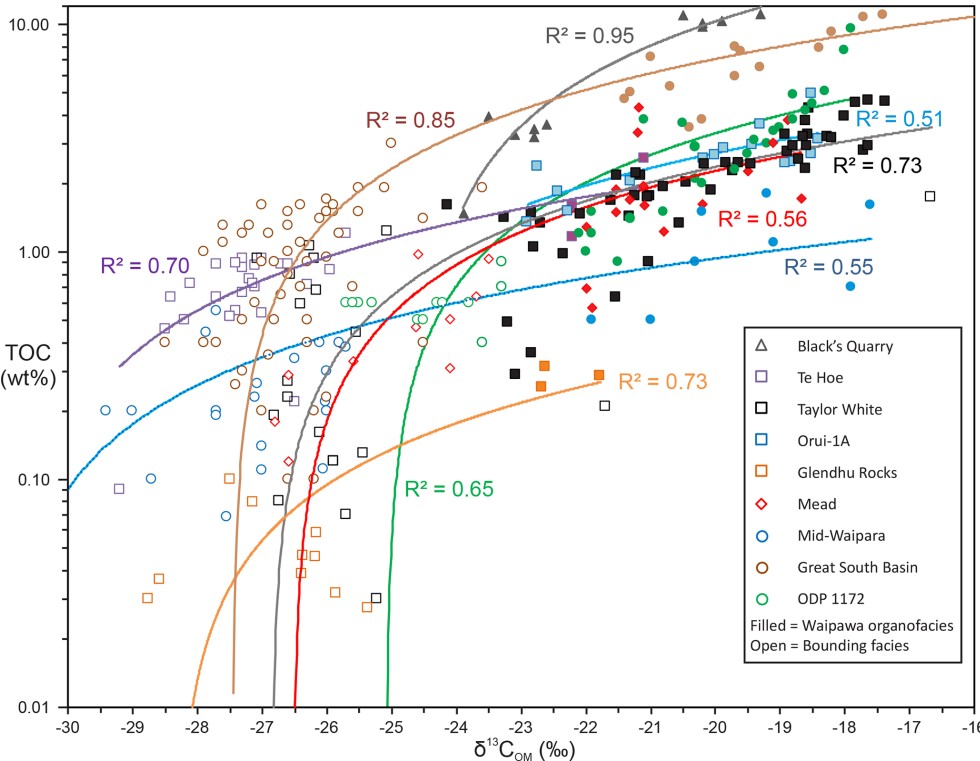

**Figure 3.** Correlation between TOC and $\delta^{13}C_{OM}$ in the studied sections. Correlation coefficients ($R^2$) relate to the linear regression lines of the same colour. Great South Basin (GSB) drill holes include Hoiho-1C, Kawau-1A, Pakaha-1 and Toroa-1.

The fatty acids from the mid-Waipara samples were extracted and analysed using a different extraction and separation protocol. Powdered samples were placed in pre-extracted cellulose thimbles and extracted under reflux using a Soxhlet apparatus for 24 h with DCM / MeOH (2 : 1 $v/v$) as the organic solvent. The resulting TLEs were separated using an aminopropyl ($NH_2$) solid phase extraction (SPE) column by elution with DCM/isopropanol (2 : 1 $v/v$; neutral fractions), followed by 2 % (by volume) acetic acid in diethyl ether (acid fractions). The columns were glass cartridges containing 500 mg of silica-bonded stationary phase, manufactured by Isolute®. Acid fractions were then methylated using a $BF_3$ / MeOH complex (14 % $w/v$; 100 μL; 60 °C for 30 min). After cooling to room temperature, $\sim 1$ mL of double-distilled water was added and then extracted with $\sim 2$ mL of DCM. The extracts were passed through a short glass pipette column packed with pre-extracted glass wool and sodium sulfate ($Na_2SO_4$) (to remove residual water). A further two repeat extractions were performed on each fraction (eluted into the same vial), resulting in a combined extract, which was then dried under $N_2$. The methylated acid fractions were then silylated at 70 °C for 1 h.

For compound-specific carbon isotope analyses of the fatty acids (Table S6) GC–C–IRMS was conducted using a Hewlett Packard 6890 gas chromatograph connected to a Thermoquest Finnigan Delta plus XL spectrometer, via a GC III combustion interface (comprising Cu, Pt and Ni wires within a fused alumina reactor at a constant temperature of 940 °C). GC conditions were as described above. Duplicate analyses were conducted for each sample, with the mass balance of values corrected for the addition of a methyl group and reported in standard delta (‰) notation relative to VPDB. Analytical precision, based on replicate analysis of a standard of mixed fatty-acid methyl esters (FAMEs), is $< \pm 0.5$‰.

## 4 Results

### 4.1 Distinguishing features of Waipawa organofacies

The Waipawa organofacies was defined by Hollis et al. (2014) as a distinctive, organic-rich marine facies in which the organic matter is enriched in $^{13}C$. The lithology is typically mudstone, with varying proportions of detrital sand, biogenic silica and glauconite. The additional samples and sites examined in this study confirm that, irrespective of background sediment type, Waipawa organofacies is readily identified by a combination of relatively elevated TOC values, typically > 1 wt % and up to about 15 wt %, and enriched $\delta^{13}C_{OM}$ values of $-24$‰ to $-17$‰ (Fig. 3, Table S2). There is a positive correlation between TOC and $\delta^{13}C_{OM}$ in all sections examined, even in sections such as Glendhu Rocks

where maximum TOC is low in comparison with other sites (i.e. $< 0.5$ wt %).

## 4.2 Terrestrial origin of $^{13}$C-enriched OM in Waipawa organofacies

### 4.2.1 Palynofacies evidence

Palynofacies analysis of Waipawa organofacies indicates that, for all sites examined, irrespective of depositional setting, facies or lithology, TOC and $^{13}$C enrichment are associated with a dominance of terrestrial OM (Table S2; Sect. S2). In samples where $\delta^{13}$C$_{OM}$ ranges from $-24$‰ to $-17$‰ (median $-20$‰), the terrestrial component of palynofacies assemblages is generally greater than 70 % (Fig. 4a). In most of these samples, the most abundant palynofacies category is degraded woody plant matter (degraded phytoclasts, Fig. 4b).

For all sections studied, total terrestrial palynodebris, total phytoclast and degraded phytoclast abundances have the strongest positive correlations with $\delta^{13}$C$_{OM}$ of all palynofacies components in bulk samples (Table S2). To investigate the relationship between $^{13}$C enrichment and palynofacies in greater detail, we carried out palynofacies and $\delta^{13}$C analyses on density-separated fractions of samples from the Taylor White section (Fig. 5, Table S3). In this analysis we differentiate between four lithofacies: (1) Whangai facies (siliceous mudstone underlying the Waipawa Formation), (2) organic-rich Waipawa facies (OM-rich; TOC $> 2$ wt %), (3) organic-poor Waipawa facies (OM-poor; $< 2$ wt %), and (4) Wanstead facies (mudstone overlying the Waipawa Formation). More detailed descriptions of the lithofacies and stratigraphy are provided by Naeher et al. (2019). In the bulk sediments, the lithofacies are readily distinguished by palynofacies: the two Waipawa facies are dominated by degraded phytoclasts, the Whangai facies has a greater proportion of marine components (Fig. 5a), and the Wanstead facies has abundant opaque phytoclasts. The dominance of opaque phytoclasts in the Wanstead facies is thought to be due to oxidation of all but the most recalcitrant carbon in fully oxygenated depositional conditions (Naeher et al., 2019). Degraded phytoclasts tend to be more abundant in the OM-rich Waipawa facies than in the OM-poor facies. The OM-rich facies also tends to have more positive $\delta^{13}$C values ($\sim -18$‰) than the OM-poor facies ($\sim -22$‰), whereas the Whangai and Wanstead facies tend to have $\delta^{13}$C values of $-26$‰ to $-27$‰ (Fig. 5b). Given the abundance of marine palynodebris in the Whangai facies (Fig. 5a), this $\delta^{13}$C range provides a baseline for marine OM prior to Waipawa deposition.

Because of the low abundance of OM in the Wanstead facies, we were not able to differentiate density fractions for this facies. For the three remaining facies, the marine component (i.e. amorphous organic matter (AOM) + marine palynomorphs (dinoflagellates)) tends to be greater in the light fraction (specific gravity (SG) $< 1.3$), and this is especially true for the Whangai and OM-poor Waipawa samples (Fig. 5c, Table S3), which likely explains the generally more depleted $\delta^{13}$C$_{OM}$ values of these two facies (Fig. 5d). For the heavy fractions (SG $> 1.3$), the terrestrial component is dominant in all three Waipawa and Whangai facies (Fig. 5e). However, degraded phytoclasts are only dominant in the fractions that are most enriched in $^{13}$C. For Whangai and some OM-poor Waipawa fractions, non-degraded or opaque phytoclasts (i.e. "Other phytoclasts" in Table S3) dominate the palynofacies assemblage. It is notable that the heavy Whangai fraction, which is dominated by terrestrial palynodebris (Fig. 5e), has an enriched $\delta^{13}$C value of $-21$‰ (Fig. 5f). This allows us to benchmark $\delta^{13}$C values prior to Waipawa deposition at $-21$‰ for terrestrial OM and $-26$‰ to $-27$‰ for marine OM (Fig. 5b, d). These values are consistent with the findings of Sluijs and Dickens (2012), who derived values of $-23.4$‰ and $-27.3$‰ for terrestrial and marine OM, respectively, for the latest Paleocene and early Eocene sediments in the Arctic Ocean. The more positive value for terrestrial OM in the New Zealand records may reflect differences in vegetation between the two regions.

A moderate, positive correlation ($R^2 = 0.56$) between degraded phytoclast abundance and $\delta^{13}$C in the terrestrial-OM-dominated heavy fractions (Fig. 5f) indicates that variation in $\delta^{13}$C in the range of $-21$‰ to $-17$‰ for Waipawa organofacies is primarily a function of the proportion of degraded phytoclasts. Conversely, more depleted $\delta^{13}$C values in the range of $-22$‰ to $-25$‰ for Waipawa organofacies appear to reflect greater contributions from marine OM sources, especially evident in the light fractions (Fig. 5c and d). It is important to note, however, that volumetrically minor contributions of marine OM are present within the heavy and light fractions of both the OM-poor and OM-rich Waipawa organofacies, with volumes reaching 26.7 % in the OM-poor samples and 8.3 % in the OM-rich samples (Table S3). Thus, despite positive correlations between $\delta^{13}$C and degraded phytoclast abundance (Fig. 5b, d, f), the subordinate contributions of marine OM will also have an influence on the $\delta^{13}$C values of bulk samples.

### 4.2.2 Geochemical evidence

Various geochemical indicators, such as higher plant biomarkers of mainly angiosperm origin, also point to the dominance of terrestrial OM in Waipawa organofacies in the Taylor White section (Naeher et al., 2019). The OM-rich Waipawa facies (Fig. 6a) is enriched in pristane, phytane and odd-carbon-numbered high molecular-weight (HMW; C$_{27}$–C$_{31}$) $n$-alkanes relative to the OM-poor Waipawa (Fig. 6b) and Whangai facies (Fig. 6c). The prevalence of HMW $n$-alkanes with odd-over-even predominance is consistent with a higher-plant-dominated organofacies (Peters et al., 2005). Although pristane and phytane are typically derived from algal chlorophyll (phytol component) in marine sediments, the dominance of terrigenous OM in Waipawa organofacies suggests that a proportion of these compounds is derived

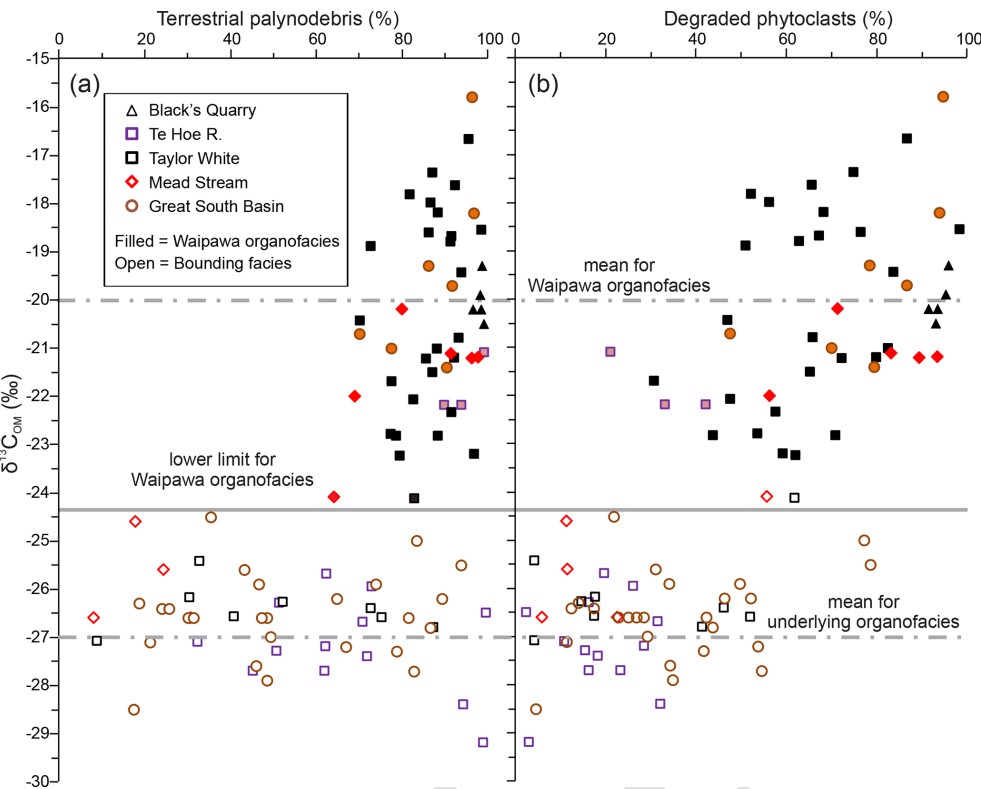

**Figure 4.** Correlation of $\delta^{13}C_{OM}$ with **(a)** terrestrial palynodebris and **(b)** degraded phytoclasts in five onshore sections and four drill holes in the Great South Basin (Toroa-1, Pakaha-1, Kawau-1A and Hoiho-1C).

from leaf chlorophyll. The same suite of compounds occurs in the OM-poor Waipawa organofacies (Fig. 6b), but the HMW $n$-alkanes are not as prominent as in the OM-rich facies (Fig. 6a), with greater proportions of low and
5 medium molecular-weight (LMW, C$_{17}$–C$_{20}$; MMW, C$_{21}$–C$_{26}$) $n$-alkanes indicating a greater marine contribution. While this facies still has a high abundance of terrestrial OM, an overall lower abundance of degraded phytoclasts explains the relatively depleted $\delta^{13}$C values compared to those of the
10 OM-rich Waipawa facies. The same suite of compounds is also present in Whangai facies (Fig. 6c) but with a greater abundance of LMW $n$-alkanes which signals an even greater marine contribution. Nonetheless, abundant HMW $n$-alkanes in all three facies indicate a significant terrestrial contribu-
15 tion throughout the succession. The persistence of LMW and MMW $n$-alkanes in Waipawa organofacies is in line with previously reported evidence for the enrichment of some specific marine biomarkers, notably C$_{30}$ steranes, which indicate that Waipawa deposition was associated with an increased
abundance of specific groups of marine algae in most settings (Murray et al., 1994; Killops et al., 2000; Hollis et al., 2014; Naeher et al., 2019).

The covariance between $\delta^{13}C_{OM}$ and terrestrial OM is further illustrated by the abundance of lignin-derived phe-
25 nols within the kerogen fraction. Naeher et al. (2019) reported a strong, positive correlation ($R^2 = 0.83$) between the

phenols / naphthalene ratio and TOC in the Taylor White section. The same relationship is evident between the phenols / naphthalene ratio and $\delta^{13}C_{OM}$ (Fig. 7a, Table S4). This correlation confirms that $\delta^{13}C_{OM}$ increases as the abundance 30 of woody material increases in the sediment.

## 4.3 Possible autogenic causes of $^{13}$C enrichment

Having established that the $^{13}$C enrichment appears to be linked primarily to the dominance of terrestrially derived degraded phytoclasts in the Waipawa OM (Fig. 5), we now 35 consider how and to what extent OM degradation or indeed preservation processes within the broader depositional environment might account for the $^{13}$C enrichment. Only by accounting for potential processes of $^{13}$C enrichment during OM transportation, deposition and early diagenesis is it pos- 40 sible to identify any residual enrichment that may be related to a drawdown in atmospheric CO$_2$ levels.

### 4.3.1 Sulfurization

The preservation of carbohydrates through sulfurization is an established mechanism for $^{13}$C enrichment in fossil organic 45 matter (Sinninghe Damsté et al., 1998; van Kaam-Peters et al., 1998) and was previously suggested for the Waipawa organofacies by Hollis et al. (2014). Naeher et al. (2019) re-

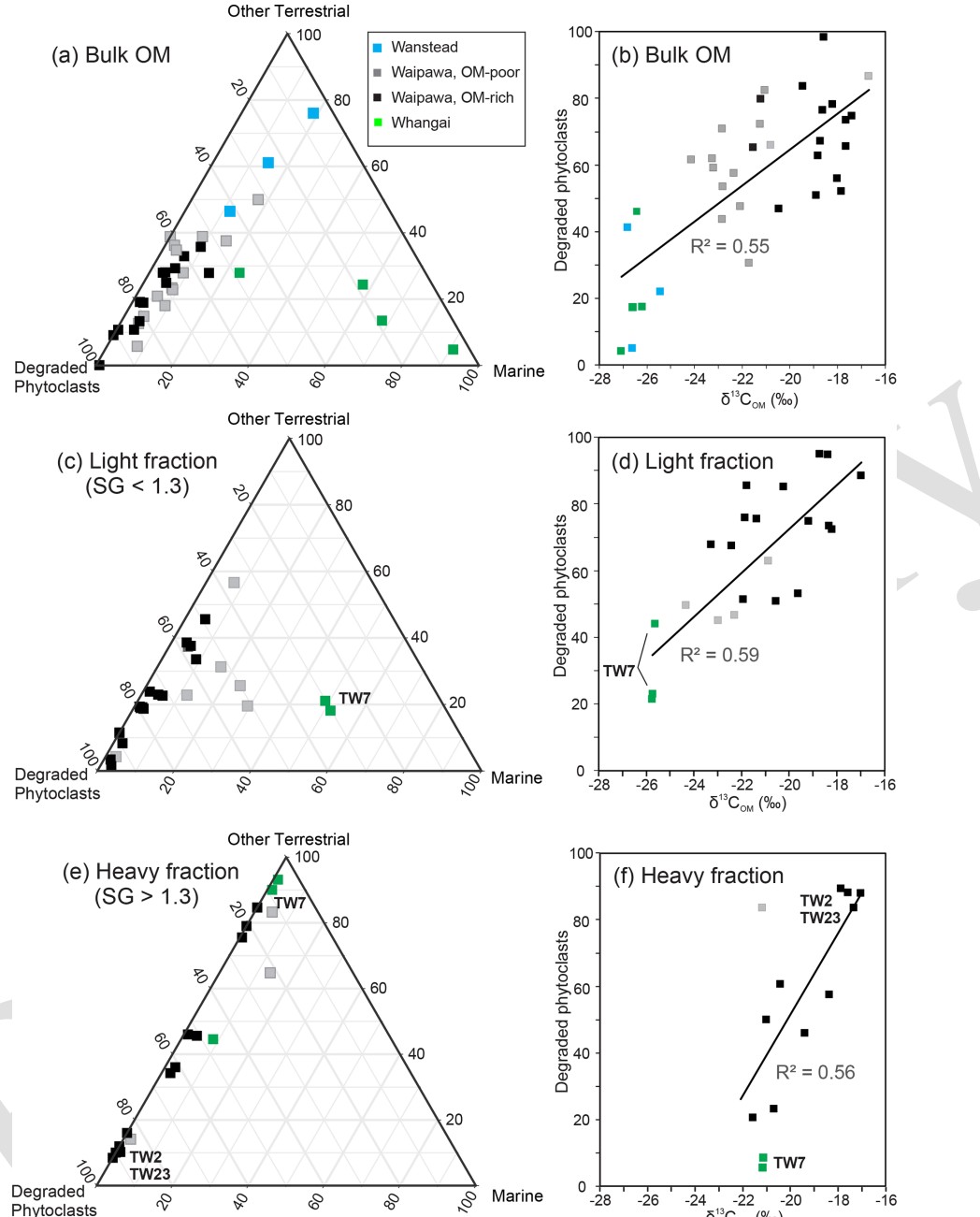

**Figure 5.** Proportions of primary palynofacies components (degraded phytoclasts, other terrestrial, marine) in **(a)** bulk organic matter and **(c)** light and **(e)** heavy density fractions compared with cross-plots of degraded phytoclasts and $\delta^{13}C_{OM}$ for **(b)** bulk organic matter and **(d)** light and **(f)** heavy density fractions for samples from four facies in the Taylor White section. Selected samples referred to in text are annotated. Density fractions were unavailable for Wanstead and some Whangai samples because of low OM contents.

ported a strong, positive correlation ($R^2 = 0.80$) between the sulfur-containing thiophenes and the hydrogen index (HI) for Waipawa and other facies in the Taylor White section. However, we find only a weak correlation ($R^2 = 0.20$) between thiophenes and $\delta^{13}C_{OM}$ for the Waipawa and Whangai samples (Fig. 7b, Table S4). This suggests that the preservation of carbohydrates by sulfurization is at most a weak secondary influence on $^{13}$C enrichment within the Waipawa organofacies. Indeed, the opposing process of tissue degradation appears to have been far more influential given the strong correlation between degraded phytoclast abundance and $\delta^{13}C_{OM}$ (Fig. 5). As noted below, degradation of plant material will also break down carbohydrate residues.

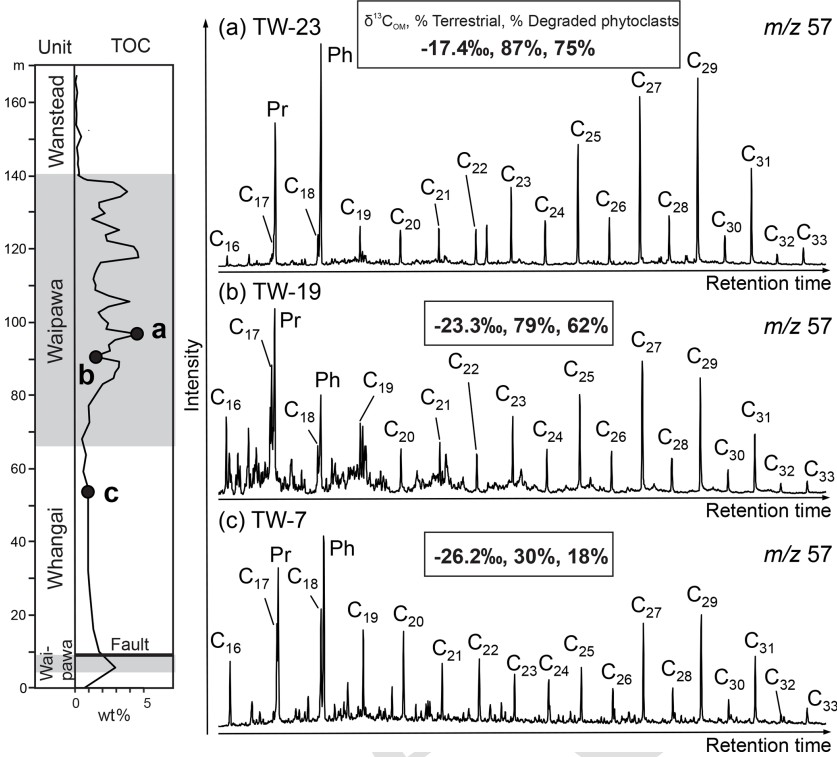

**Figure 6.** Representative biomarker chromatograms for **(a)** organic-rich Waipawa facies, **(b)** organic-poor Waipawa facies and **(c)** Whangai facies in the Taylor White section. The $\delta^{13}C_{OM}$ value and percentages of terrestrial palynodebris and degraded phytoclasts are also shown for each sample. Identified peaks are $n$-alkanes, pristane (Pr) and phytane (Ph).

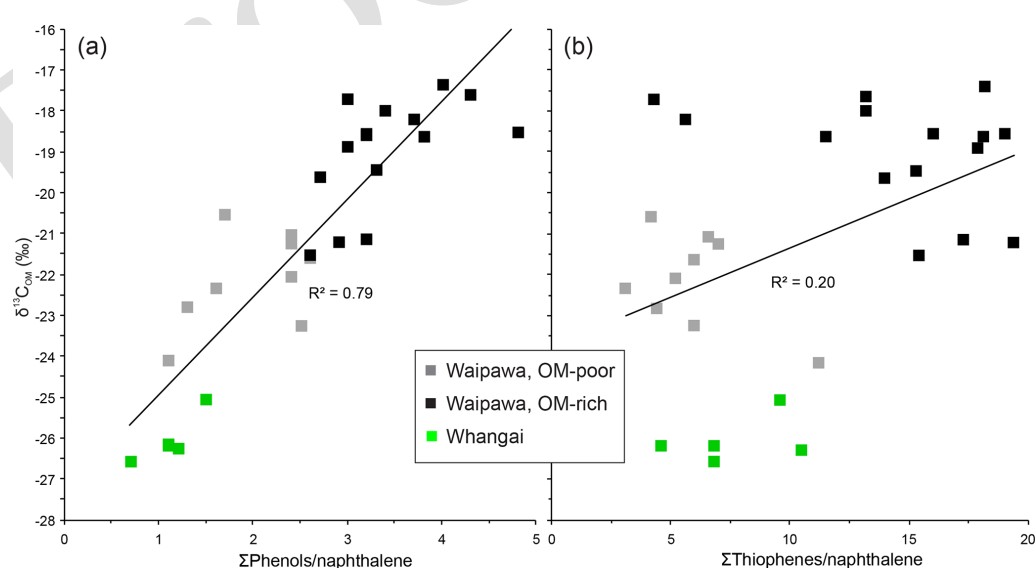

**Figure 7.** Correlation of $\delta^{13}C_{OM}$ with **(a)** phenol and **(b)** thiophene abundances relative to naphthalene for the OM-rich (TOC > 2 wt %) and OM-poor (TOC < 2 wt %) Waipawa facies and Whangai facies in the Taylor White section. Naphthalene is used to normalize these compounds because it is a generic compound independent of source. Linear regressions include all samples.

### 4.3.2 Lignin degradation

The main macromolecular components of woody plants are cellulose and lignin. These components have different susceptibilities to degradation, and this has been shown to alter the $\delta^{13}$C of woody plant matter (Gröcke et al., 1999; Schleser et al., 1999; Schweizer et al., 1999; van Bergen and Poole, 2002; Fernandez et al., 2003). According to these studies, $\delta^{13}C_{OM}$ decreases during early diagenesis due to the rapid degradation of cellulose and the consequent increase in the relative concentration of the more recalcitrant lignin. However, eventually the lignin begins to degrade too, and this can lead to an increase in $\delta^{13}C_{OM}$ values (Hedges et al., 1985; Gröcke, 1998; Gröcke et al., 1999; van Bergen and Poole, 2002). Van Bergen and Poole (2002, their Fig. 1) suggest enrichment in $\delta^{13}$C of up to 7‰–8‰ can occur in lignin as a result of demethylation and hydroxylation reactions. This seems a plausible cause for the relationship described above for the heavy fraction (SG < 1.3) of the OM-rich Waipawa facies samples from the Taylor White section (Fig. 5f) in which an increase in the proportion of degraded phytoclasts from 20 % to 90 % corresponds to a ∼ 5‰ increase in $\delta^{13}$C. The heavy fraction provides the best guide to the effects of phytoclast degradation on $\delta^{13}$C values because this fraction contains less marine organic matter than the light fraction (Fig. 5c, e).

For this study, the degree of $^{13}$C enrichment due to lignin degradation can be gauged by considering the $\delta^{13}$C variation in the aromatic and saturated hydrocarbon fractions. Aromatic compounds are rare in marine organisms but abundant in the lignin of land plants (Sofer, 1984). If lignin degradation is the main cause of $^{13}$C enrichment within the Waipawa organofacies, we would expect the aromatic hydrocarbon fraction to be more enriched in $^{13}$C than the saturated fraction. This is indeed what we observe for OM-rich Waipawa organofacies (Figs. 8, 9a). However, this relationship is partly explained by the covariance between $\delta^{13}C_{OM}$ and terrestrial OM because the aromatic fraction of terrestrial OM is typically enriched in $^{13}$C relative to the saturated fraction (Sofer, 1984). A study of Paleocene coal and coaly mudstone samples (Sykes and Zink, 2012) indicates that the typical difference between aromatic and saturated $\delta^{13}$C in Paleocene terrestrial OM is ∼ 2‰–3‰ (Fig. 8, Table S5). The difference between OM-rich Waipawa samples and the underlying Whangai facies is ∼ 4‰–5‰, which implies that ∼ 2‰ of the difference between $\delta^{13}C_{Aro}$ and $\delta^{13}C_{Sat}$ may be due to lignin degradation. We conclude that the difference in the positive carbon isotope excursion (CIE) between aromatic and saturated hydrocarbons can be attributed to the dominance of degraded terrestrial OM. The positive CIE of ∼ 1‰ recorded in saturated hydrocarbons (Table S5) may be a better guide to changes in $\delta^{13}$C within the exogenic carbon cycle, although this value may also be affected by the mixing of marine and terrestrial sources.

### 4.4 $^{13}$C enrichment attributable to changes in the exogenic carbon cycle

To circumvent the complications of lignin degradation and source mixing that affect the $\delta^{13}$C values of the bulk OM and the saturated and aromatic hydrocarbon fractions, we now focus on compound-specific carbon isotope analysis of sediments from the Taylor White (Fig. 9a, b) and mid-Waipara sections (Fig. 10a). For Taylor White, we analysed pristane, phytane and C₁₈–C₃₃ $n$-alkanes ($\delta^{13}C_{Pr}$, $\delta^{13}C_{Ph}$, $\delta^{13}C_{18}$, etc.; Table S5). For mid-Waipara, we analysed the C₁₆–C₃₂ $n$-alkanoic or fatty acids (Table S6) because there were insufficient concentrations of $n$-alkanes in the OM from this section. For most compounds in both sections, irrespective of whether they are derived from terrestrial, aquatic or marine sources, the $\delta^{13}$C trends parallel that of the bulk OM to a greater or lesser extent (Figs. 9, 10; Tables S5, S6; Sect. S4).

Despite parallel trends, the magnitudes of the positive CIE differ considerably (Fig. 11). Of the compound-specific components, phytane exhibits the largest CIE (∼ 7‰), followed by pristane and the fatty acids (∼ 3‰–5‰), the LMW and MMW $n$-alkanes (2‰–3‰), and then the HMW $n$-alkanes (∼ 1‰). Unless noted otherwise, the CIEs referred to here are the difference between the mean of samples from OM-rich Waipawa organofacies and the mean of samples from the underlying facies (Tables S5, S6).

Variations in the magnitude of the CIEs amongst different compound classes are commonly due to mixing of OM sources, as discussed above for bulk fractions, or due to different isotopic sensitivities to environmental change in the source organisms (Pancost et al., 1999; Schouten et al., 2007). A strong correlation between pristane and phytane carbon isotope compositions and $\delta^{13}C_{OM}$ (Table S5) suggests that they derive from a mixture of terrestrial and aquatic sources in the Taylor White section, with the terrestrial source dominant in OM-rich Waipawa organofacies. A positive CIE for pristane indicates that the primary terrestrial substrate may be enriched in $^{13}$C by ∼ 4‰ (Fig. 11c). The greater CIE for phytane suggests an additional unidentified source of enriched carbon.

The LMW and MMW $n$-alkanes and fatty acids, which are thought to be derived mainly from aquatic sources, exhibit positive CIEs of 2‰–5‰ (Fig. 11c, d). These shifts may reflect a change in substrate $\delta^{13}$C or secondary processes that enhance carbon isotope discrimination during photosynthesis such as a decline in atmospheric CO₂ concentration or an increase in plant growth rates (Bidigare et al., 1997).

For the higher plant biomarkers, HMW $n$-alkanes at Taylor White exhibit a much lower CIE (∼ 1‰) than the HMW fatty acids at mid-Waipara (∼ 3‰). It seems that the small magnitude of the CIE for HMW $n$-alkanes at Taylor White may reflect some mixing of terrestrial OM sources. A combination of contemporaneous and older reworked sources was found to have dampened the signal from contemporaneous higher plants in a study of the Paleocene–Eocene tran-

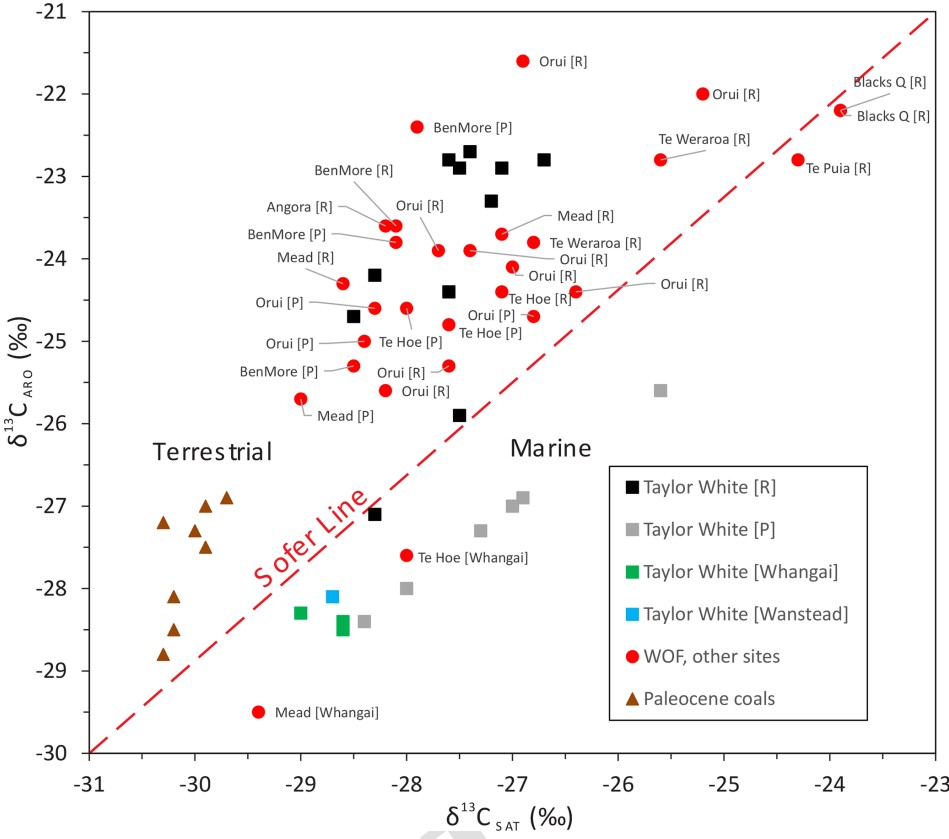

**Figure 8.** Relationship between $\delta^{13}$C values for aromatic (ARO) and saturated (SAT) hydrocarbon fractions for the OM-rich (TOC > 2 wt %, [R]) and OM-poor (TOC < 2 wt %, [P]) Waipawa organofacies and underlying Whangai facies shown as a cross-plot of the two variables. Samples of Paleocene coaly rocks are included for comparison (see Table S5 for sample details). The Sofer line is used to separate marine and terrestrial OM sources in oils (Sofer, 1984).

sition in Tanzania (e.g. Carmichael et al., 2017). We find support for this possibility in additional compound-specific $\delta^{13}$C analyses undertaken on OM-rich Waipawa organofacies at Angora Road and Mead Stream (Fig. 12; Table S5). At these sites, Waipawa organofacies has similar $\delta^{13}$C values for phytane, pristane, LMW and MMW $n$-alkanes as at Taylor White. However, in contrast to the weakly enriched HMW $n$-alkane values at Taylor White, the HMW $n$-alkanes at Angora Stream and Mead Stream are as enriched as the shorter $n$-alkanes at these sites. This suggests that the relatively depleted HMW values at Taylor White may be due to mixing with an older source and the contemporaneous higher plant input is better represented by values of $\sim -27.5$ ‰ recorded at Mead and Angora. This value indicates an excursion of $\sim 3$ ‰, which is in line with the HMW fatty acids at mid-Waipara.

In summary, most of the organic compounds we have analysed are enriched in $^{13}$C in the OM-rich Waipawa organofacies relative to bounding facies. The evidence indicates a significant excursion of at least 2 ‰ for compounds derived from both aquatic and terrestrial OM. Crucially, we have demonstrated that the $\delta^{13}$C trends in the aquatic and terrestrial biomarkers parallel trends in bulk OM (Figs. 9, 10). This implies that the primary influence on $\delta^{13}C_{OM}$, the proportion of degraded woody plant matter, is modulated by the same carbon cycle changes that cause the variation in $\delta^{13}$C in aquatic and terrestrial biomarkers.

## 4.5 Correlation and age of Waipawa organofacies deposition

In order to establish how Waipawa deposition might be linked to global climate, we reviewed and revised the age control for Waipawa organofacies. Based on a combination of biostratigraphy (Schiøler et al., 2010; Crouch et al., 2014; Kulhanek et al., 2015) and limited magnetostratigraphy, Hollis et al. (2014) inferred that Waipawa organofacies deposition occurred over $\sim 700$ kyr between $\sim 59.4$ and $\sim 58.7$ Ma (GTS2012, Gradstein et al., 2012). We have extended the bulk carbonate $\delta^{13}$C stratigraphy for the Paleogene section at Mead Stream (Hollis et al., 2005; Nicolo et al., 2007; Slotnick et al., 2012) to encompass the interval spanning the Waipawa organofacies (Fig. 13; Sect. S3). Mead Stream has the most complete known record of Waipawa organofacies

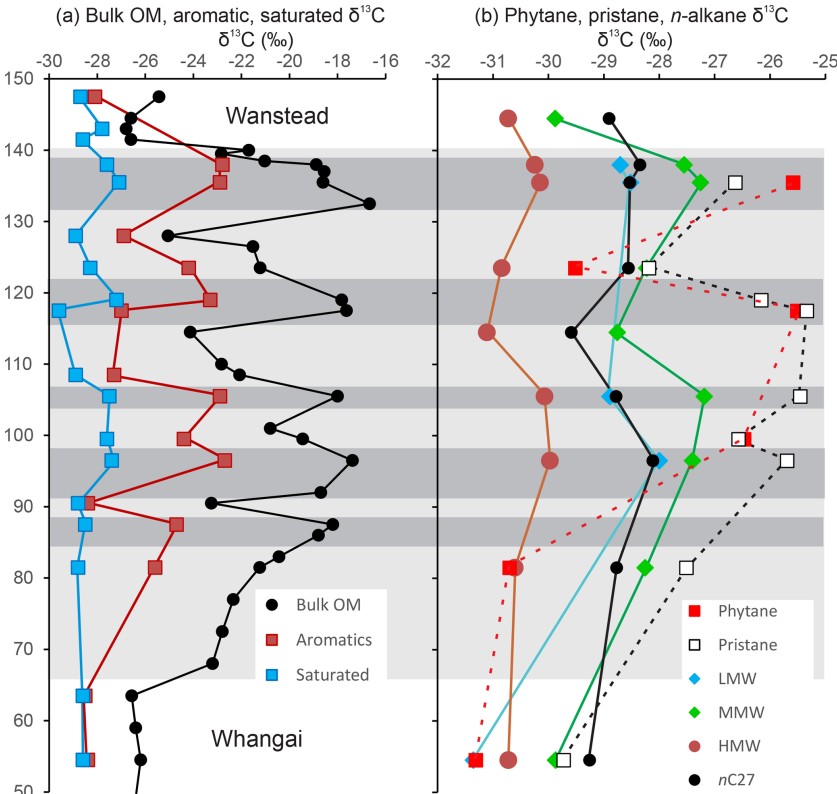

**Figure 9.** Stratigraphic and facies-related variation in **(a)** bulk organic, total aromatic, total saturated and **(b)** compound-specific $\delta^{13}$C values in the Taylor White section. Average values for odd-numbered LMW ($n$C$_{19}$), MMW ($n$C$_{21}$–$n$C$_{25}$) and HMW ($n$C$_{27}$–$n$C$_{33}$) $n$-alkanes in addition to a representative HMW $n$-alkane ($n$C$_{27}$). Horizontal bands represent OM-poor (pale grey) and OM-rich (medium grey) Waipawa organofacies.

in a distal setting (Fig. S5a), with a gradational base and sharp but potentially conformable top (Fig. S5). It also has a bulk carbonate $\delta^{13}$C record that parallels benthic and bulk carbonate records in deep-sea sediment cores (Hollis et al., 2005; Nicolo et al., 2007; Slotnick et al., 2012). This allows us to correlate the interval to high-resolution stable isotope records from North Pacific ODP Site 1209 and South Atlantic ODP Site 1262 (Westerhold et al., 2008, 2011, 2020; Littler et al., 2014; Barnet et al., 2019). We use the 2020 Geological Timescale for the Paleogene (Speijer et al., 2020), which incorporates the astronomical age control of Westerhold et al. (2008, 2011, 2017, 2020).

Five CIEs in the Paleocene bulk carbonate $\delta^{13}$C record form the basis of an age model that encompasses Waipawa organofacies deposition (Fig. 13): the early late Paleocene CE4 event (ELPE), 59.3 Ma; the unnamed "*" event of Littler et al. (2014) and Barnet et al. (2019), 58.15 Ma; the B2 CIE, 57.25 Ma; the C2 CIE, 56.85 Ma; and the PETM, 55.93 Ma (Fig. 13a). Although alternative correlations are possible, this age model is in best agreement with biostratigraphy for the section (Hollis et al., 2005) and age control from other sections (Hollis et al., 2014), does not require major changes in sediment accumulation rate within uni-

form lithofacies, and retains a consistent relationship with the record from ODP Site 1262 at which Mead Stream $\delta^{13}$C values tend to be slightly more depleted. This age model provides a revised age estimate for the main phase of Waipawa organofacies of 59.2–58.5 Ma, which is the same duration but slightly younger than previous estimates (Hollis et al., 2014) and consistent with a new age estimate of $57.5 \pm 3.5$ Ma derived from Re–Os geochronology (Rotich et al., 2020). In contrast to sections where sedimentation rates increase during Waipawa deposition, carbonate flux decreases in the pelagic setting and sedimentation rate decreases markedly (Hollis et al., 2014). A second 20 cm thick layer of Waipawa organofacies occurs 5 m above the main phase of deposition. Our age model suggests this layer is correlated with the "*" event. Both this CIE and the ELPE are possible hyperthermals but have inconsistent evidence for warming (Littler et al., 2014; Barnet et al., 2019). In contrast to the Eocene CIEs at Mead Stream, which are associated with marl-rich intervals in the micritic limestone succession (Hollis et al., 2005; Slotnick et al., 2012), no obvious changes in lithology are associated with the ELPE and the "*" CIE is centred on a thin siliceous Waipawa organofacies layer.

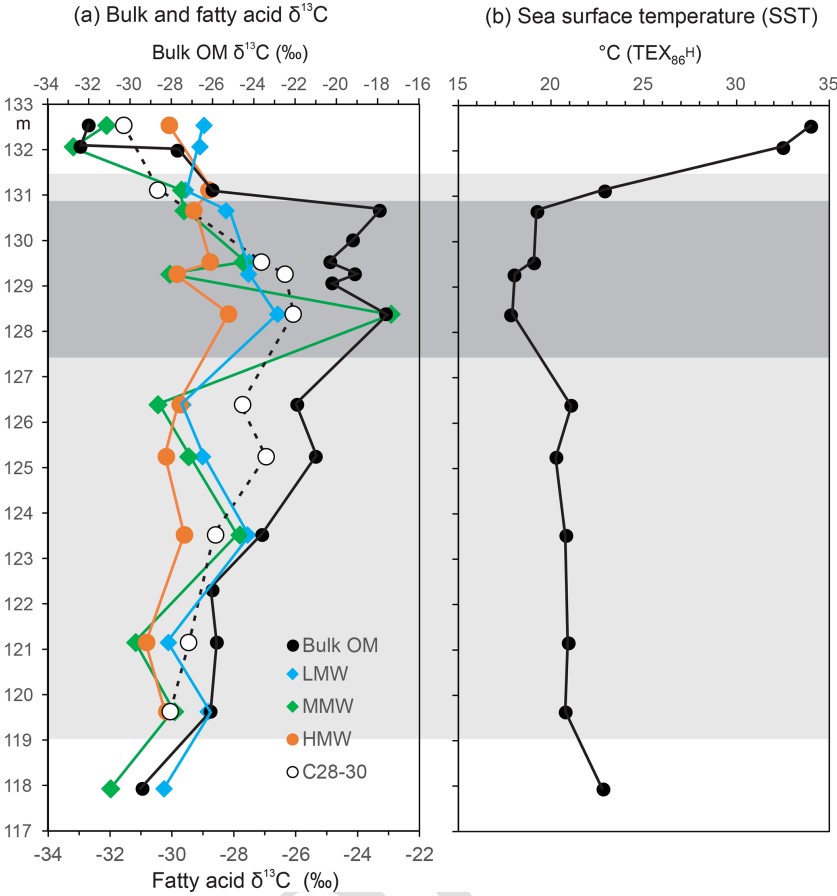

**Figure 10.** Stratigraphic and facies-related variation in **(a)** bulk organic and compound-specific $\delta^{13}C$ values and **(b)** sea surface temperature in the mid-Waipara section. Average values are plotted for even-numbered LMW ($C_{16}$, $C_{18}$), MMW ($C_{20}$–$C_{26}$) and HMW ($C_{28}$–$C_{32}$) fatty acids, in addition to the mean for $C_{28}$–$C_{30}$ fatty acids. Horizontal bands represent OM-poor (pale grey) and OM-rich (medium grey) Waipawa organofacies.

The main phase of Waipawa deposition spans the first 700 kyr of the Paleocene carbon isotope maximum (PCIM; Fig. 13b), a $\sim 1.9$ Myr episode in which $\delta^{13}C$ values for marine carbonate reach their Cenozoic maximum. The positive CIE associated with the PCIM is $\sim 1$‰ in benthic foraminifera and in bulk carbonate and therefore accounts for only a small portion of the residual Waipawa organofacies CIE of 2‰–3‰. Waipawa deposition also occurs within a Paleogene maximum in marine carbonate $\delta^{18}O$, which extends from 59.6 to 58.2 Ma (Figs. 1, 13c). We refer to this interval here as the Paleocene oxygen isotope maximum (POIM). The positive shift in benthic $\delta^{18}O$ signals cooling of bottom waters despite the interval including the two possible hyperthermals noted above: the ELPE and the "∗" event, the latter coinciding with the termination of the POIM (Fig. 13a, b). In addition, Waipawa deposition coincides with a marked decrease in the coarse fraction in foraminiferal residues from ODP sites 1209 and 1262 (Fig. 1). This represents a marked decrease in the abundance of planktic foraminifera and is attributed to carbonate dissolution (Littler et al., 2014). Al-

though short-lived dissolution episodes have been linked to the PETM and other early Eocene hyperthermals (Zachos et al., 2005; Alexander et al., 2015), the longer duration of this episode and its association with positive shifts in both $\delta^{13}C$ and $\delta^{18}O$ suggest a link to climatic cooling and carbon burial (Hilting et al., 2008).

## 4.6 Waipawa organofacies associated with climatic cooling and $CO_2$ drawdown

Covariance between $\delta^{13}C_{OM}$ and the $TEX_{86}$ sea surface temperature (SST) proxy at mid-Waipara River and ODP Site 1172 (Fig. 10b, Sect. S4) indicates that Waipawa deposition is associated with regional cooling of coastal waters (Hollis et al., 2014; Bijl et al., 2021). Regional cooling on land is also indicated by temperature reconstructions based on pollen assemblages at Site 1172 (Contreras et al., 2014). Correlation with the POIM further suggests that the positive $\delta^{13}C$ excursion in Waipawa organofacies (Fig. 13c–f) is linked to global cooling of $\sim 1$ °C. Regional SST decreased

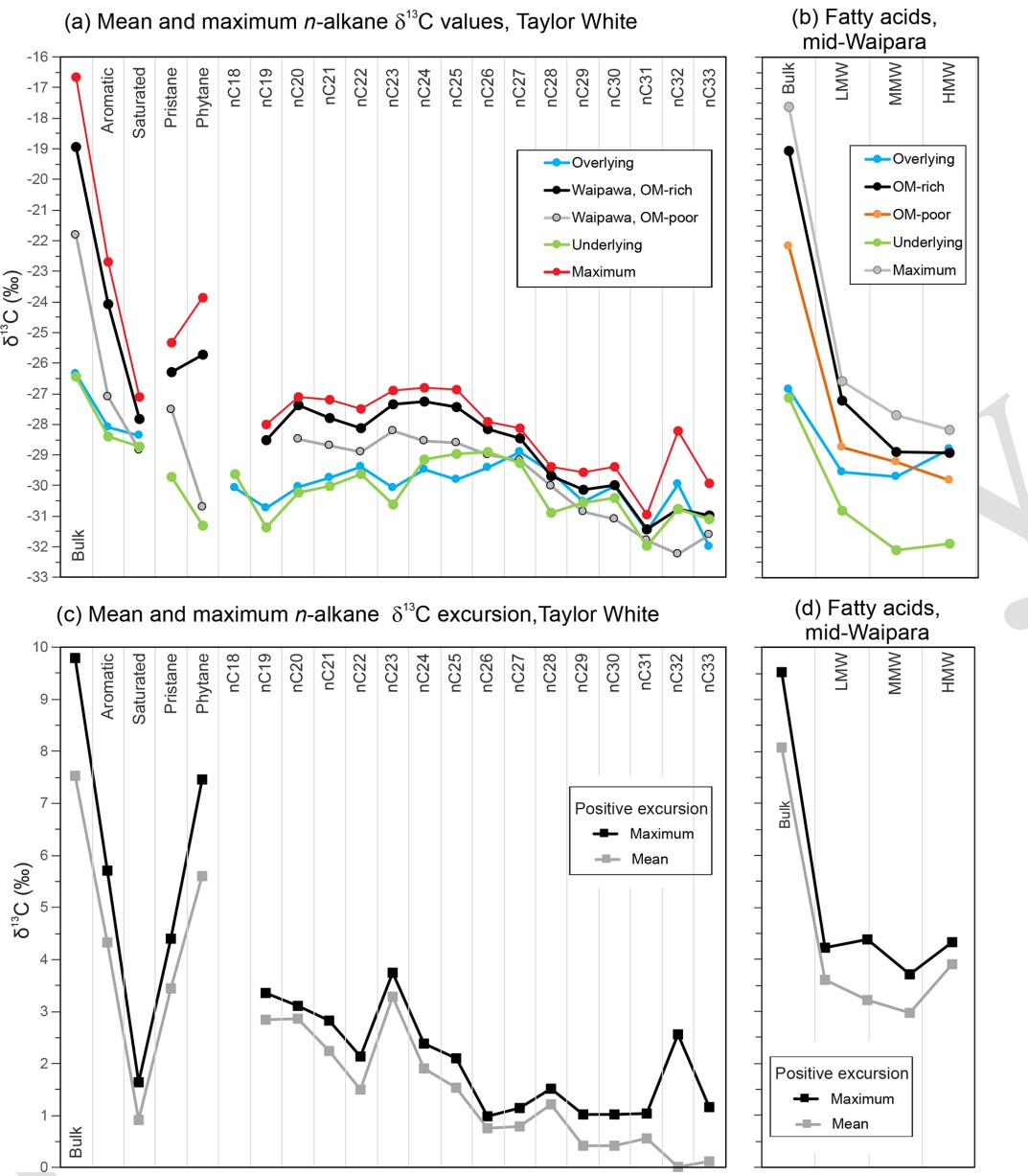

**Figure 11.** Mean compound group and compound-specific $\delta^{13}C$ values for the four facies (underlying, overlying, OM-rich and OM-poor Waipawa) and maximum values for Waipawa organofacies in the **(a)** Taylor White and **(b)** mid-Waipara sections and mean and maximum positive $\delta^{13}C$ excursion values for the **(c)** Taylor White and **(d)** mid-Waipara sections. The excursion values were derived by subtracting OM-rich Waipawa mean and maximum values from mean values for the underlying facies.

by 4–6 °C, but this might reflect localized phenomena such as enhanced upwelling of Antarctic deep water (Hollis et al., 2014).

Carbon isotope shifts of $\sim 2\,\%_o$–$3\,\%_o$ in marine and terrestrial biomarkers may be caused by a range of environmental factors, but a correlation with climatic cooling suggests that this positive CIE may be due to a decline in atmospheric $CO_2$. A decline in $CO_2$ will result in $^{13}C$ enrichment in the biomass of algae (Freeman and Hayes, 1992) as well as in higher plants (Schubert and Jahren, 2012). Most other en-

vironmental factors, such as changes in lapse rate (Körner et al., 1988), would not affect terrestrial and marine $\delta^{13}C$ to similar degrees.

We have used the relationship between atmospheric $CO_2$ and $C_3$ plant tissue $\delta^{13}C$ values (Cui and Schubert, 2016, 2017, 2018; Schubert and Jahren, 2012, 2018) to estimate atmospheric $CO_2$ concentrations prior to and during Waipawa deposition (Fig. 14). The change in $\delta^{13}C$ ($\Delta^{13}C$) per part per million of $CO_2$ follows a hyperbolic relationship (Schubert and Jahren, 2012) and is based on the model of carbon iso-

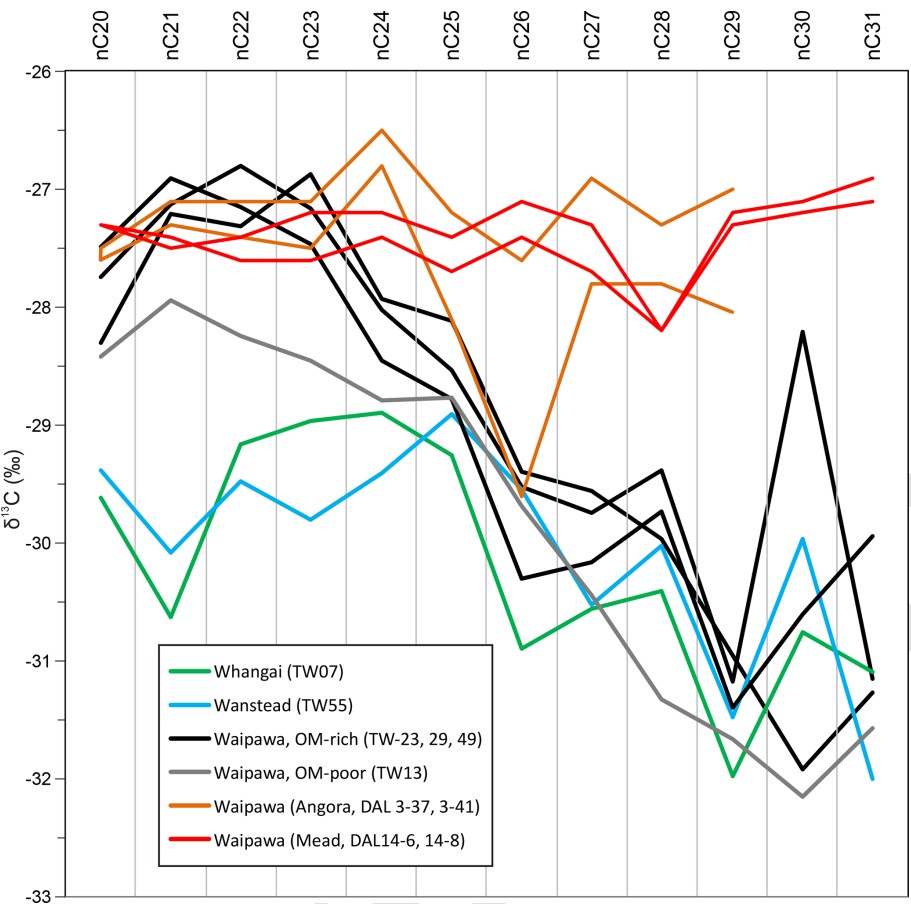

**Figure 12.** $C_{20}$–$C_{31}$ *n*-alkane $\delta^{13}C$ profiles for representative samples from four marine organofacies (Wanstead, Whangai, OM-rich and OM-poor Waipawa) in the Taylor White section compared to Waipawa organofacies in the Angora Road and Mead Stream sections.

tope fractionation in plants originally described by Farquhar et al. (1989). This proxy yields an estimate for $CO_2$ that is based on the relative change in $\Delta^{13}C$ between the time of interest ($\Delta^{13}C_{(t)}$) and the $\Delta^{13}C$ value at a chosen initial time ($\Delta^{13}C_{(t=0)}$), which is designated as $\Delta(\Delta^{13}C)$ and expressed as Eq. (1):

$$\Delta\left(\Delta^{13}C\right) = \frac{\left[(A)(B)\left(CO_{2(t)}+C\right)\right]}{\left[A+(B)\left(CO_{2(t)}+C\right)\right]} - \frac{\left[(A)(B)\left(CO_{2(t=0)}+C\right)\right]}{\left[A+(B)\left(CO_{2(t=0)}+C\right)\right]}, \quad (1)$$

where $A$, $B$ and $C$ are curve fitting parameters and solved for $CO_2$ at any time $t$ ($CO_{2(t)}$) using Eq. (2) (Cui and Schubert, 2016):

$$CO_{2(t)} = \frac{\begin{bmatrix} \Delta\left(\Delta^{13}C\right)\cdot A^2 + \Delta\left(\Delta^{13}C\right)\cdot A\cdot B\cdot CO_{2(t=0)} \\ +2\cdot\Delta\left(\Delta^{13}C\right)\cdot A\cdot B\cdot C + \Delta\left(\Delta^{13}C\right)\cdot B^2 \\ \cdot C\cdot CO_{2(t=0)} + \Delta\left(\Delta^{13}C\right)\cdot B^2\cdot C^2 \\ +A^2\cdot B\cdot CO_{2(t=0)} \end{bmatrix}}{\begin{bmatrix} A^2\cdot B - \Delta\left(\Delta^{13}C\right)\cdot A\cdot B - \Delta\left(\Delta^{13}C\right) \\ \cdot B^2\cdot CO_{2(t=0)} - \Delta\left(\Delta^{13}C\right)\cdot B^2\cdot C \end{bmatrix}}. \quad (2)$$

The combined uncertainty of parameters used to derived the estimate for atmospheric $CO_2$ is relatively large and increases with increasing $CO_2$ (Cui and Schubert, 2016, 2018).

As in Cui and Schubert (2018), we use the latest Paleocene (56.1–56.5 Ma, $t = 0$) as the reference time and adopt the same parameters (Table 1) with some modifications. We exclude an unusually low estimate of 100 ppm for $CO_2$ derived from paleosols by Sinha and Stott (1994), and we base our estimates for the $\delta^{13}C$ of atmospheric $CO_2$ ($\delta^{13}C_{CO_2}$; Fig. 14d) on the method described by Tipple et al. (2010) but recalculated using the smoothed LOESS benthic foraminiferal $\delta^{13}C$ and $\delta^{18}O$ compilation of Westerhold et al. (2020). For this calculation, we use the temperature equation of Kim and O'Neil (1997) rather than that of Erez and Luz (1983), which was developed for planktonic foraminifera and is not appropriate for benthic foraminiferal calcite (Hollis et al., 2019). We assume ice-free conditions for this calculation (i.e. $\delta^{18}O_{seawater} = -1\,‰$), while noting that the findings of this study imply the growth of ice sheets during Paleocene episodes. The three time slices used for our $\delta^{13}C_{CO_2}$ reconstructions are the latest Paleocene (pre-PETM) reference time slice (56–56.2 Ma), Waipawa organofacies (WOF;

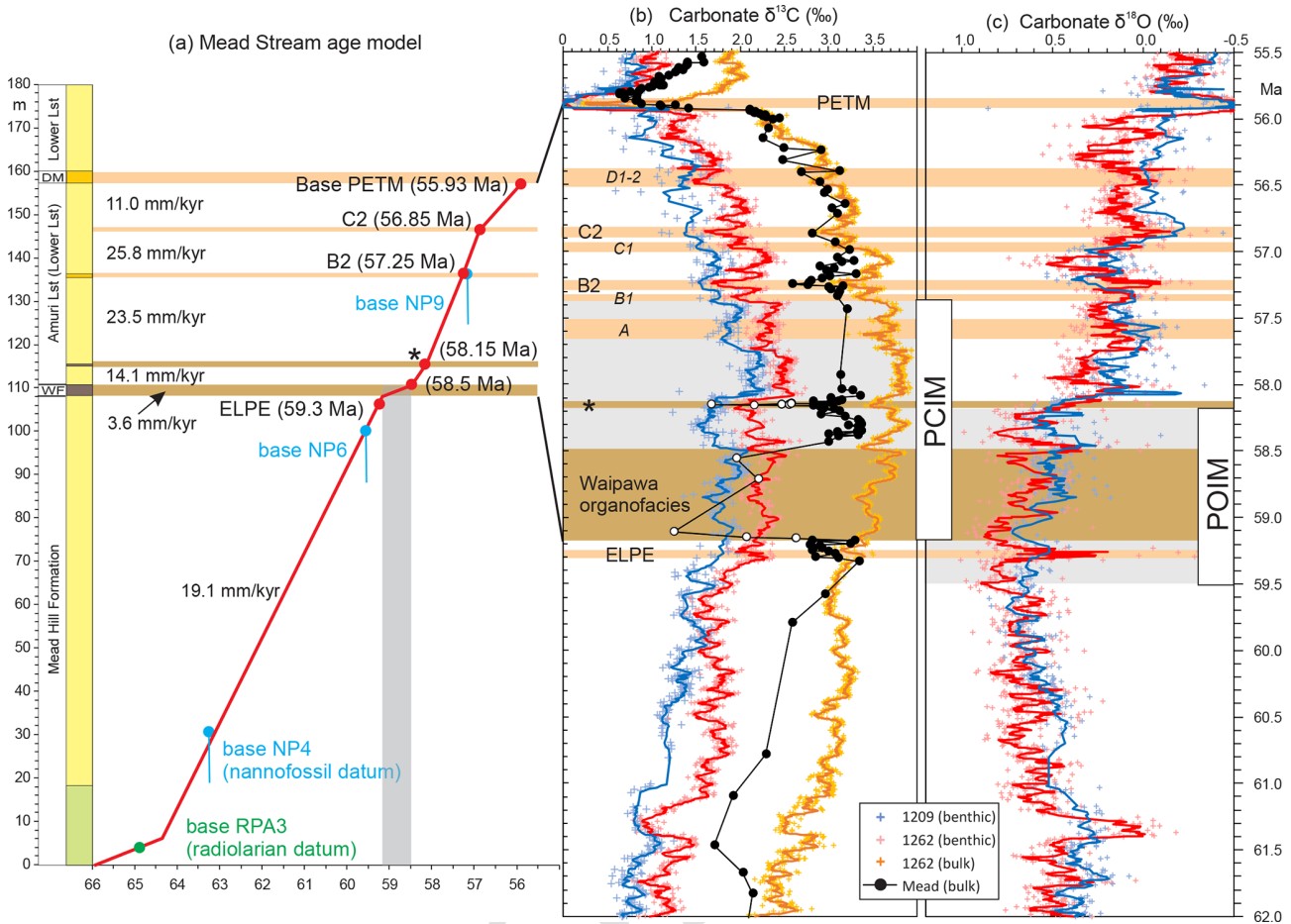

**Figure 13.** Revised age range for Waipawa organofacies based on **(a)** an age model for Paleocene sediments at Mead Stream (biostratigraphy after Hollis et al., 2020) and **(b)** correlation with high-resolution bulk carbonate and benthic foraminiferal **(b)** $\delta^{13}$C and **(c)** $\delta^{18}$O records from North Pacific ODP Site 1209 and South Atlantic ODP Site 1262 (Barnet et al., 2019; Westerhold et al., 2020). Stable isotope correlation of the Paleocene section at Mead Stream utilizes five carbon isotope excursions (CIEs) as tie points: ELPE, the unnamed "*" event at 58.15 Ma, B2, C2 and the PETM (Table S7). White filled circles in the Mead $\delta^{13}$C record are considered unreliable due to low carbonate content (< 5 wt %). Isotope curves for ODP sites are five-point moving averages.

58–59.2 Ma) and underlying organofacies (pre-WOF; 59.6–59.8 Ma) (Table 1, Fig. 14).

We have derived three estimates for the change in CO₂ that can be linked to Waipawa deposition. These are based on estimated bulk terrestrial $\delta^{13}$C values as well as $\delta^{13}$C values for the higher plant biomarkers, odd-numbered HMW $n$-alkanes ($C_{27}$–$C_{33}$) and even-numbered HMW fatty acids ($C_{26}$–$C_{32}$). CE5 For the lipid biomarker calculations, we add 4‰ to the raw $\delta^{13}$C values to account for isotope effects during the biosynthesis of $n$-alkyl biomolecules (Diefendorf et al., 2015). Similarly, in the absence of equivalent $n$-alkane and fatty-acid data for the latest Paleocene, we subtract 4‰ from the terrestrial reference value, which is derived from a latest PETM coal deposit in northeast China (Chen et al., 2014).

For HMW fatty acids in the mid-Waipara section, the carbon isotope excursion (CIE) from the mean value for under-lying facies to the mean value for the main phase of Waipawa deposition is 2.6‰ (Table S6, mean values of −31.6‰ and −29‰). For HMW $n$-alkanes in the Taylor White section, we have argued that the HMW $n$-alkanes in the Waipawa facies have been affected by mixing. If we substitute values from the nearby Angora Road section (Table S5), we derive a CIE of 3.3‰ based on the average of two OM-rich Waipawa samples from Angora Road (value of −27.9‰) and a single sample from underlying Whangai facies in the Taylor White section (value of −30.7). Because we cannot be sure of the extent to which the bulk terrestrial $\delta^{13}$C values are affected by lignin alteration, we have adopted an intermediate value of 3‰ for the bulk organic CIE. We use the $\delta^{13}$C values from the density fractions from the Taylor White section (Sect. 5.2.1) to derive a value of −21‰ for terrestrial OM in underlying Whangai facies. A CIE of 3‰ implies a value of −18‰ for Waipawa organofacies. As the

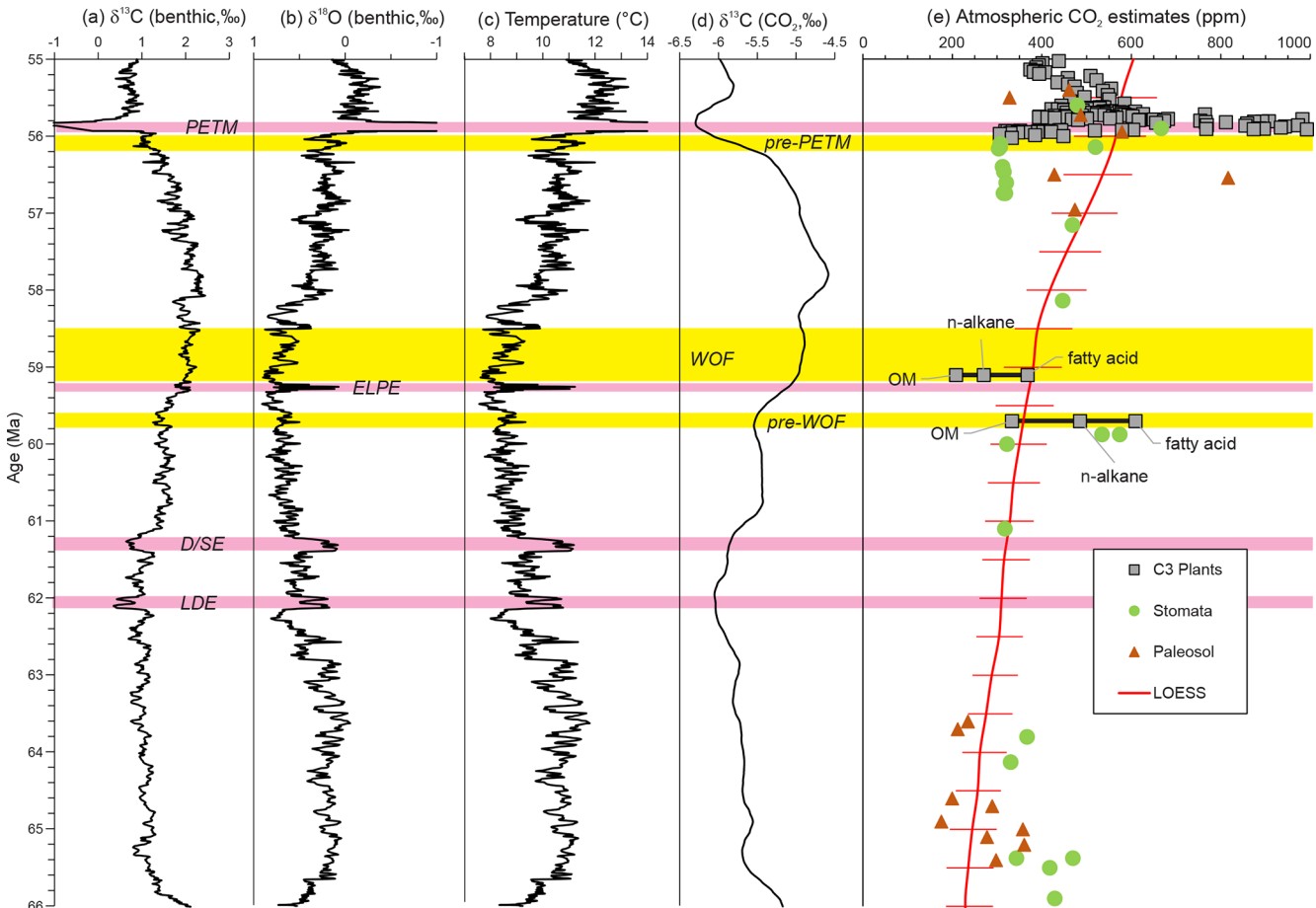

**Figure 14.** Compilation of early Paleogene variation in deep-sea benthic foraminiferal **(a)** carbon and **(b)** oxygen isotopes (LOESS smoothed curves from Westerhold et al., 2020), **(c)** oxygen isotope-based temperatures, **(d)** carbon isotope values for atmospheric CO$_2$ and **(e)** estimates for atmospheric CO$_2$ volume (after Foster et al., 2017; Hollis et al., 2019; LOESS curve from Foster et al., 2017). Horizontal pink bands – hyperthermals or carbon isotope excursion; horizontal yellow bands – reference time slices for CO$_2$ determinations; WOF indicates Waipawa organofacies. TS5

maximum value for Waipawa organofacies is −16.7‰, this suggests that lignin degradation may only account for ~1‰ of the total excursion.

The three approaches result in significant differences in CO$_2$ estimates, both for Waipawa facies and the underlying facies (Table 1). CO$_2$ estimates range from 208 to 368 ppm for Waipawa organofacies and from 333 to 609 ppm for the underlying facies. This represents a 37%–44% decrease in CO$_2$ during Waipawa deposition. This variation in values is to be expected given the many sources of uncertainty related to estimating the magnitudes of the CIEs for each parameter, variability within biomarkers and uncertainties in the calibration itself. CE6 Nevertheless, the different approaches yield consistent estimates of a ~40% decrease in CO$_2$ that can be linked to Waipawa deposition. Temperature estimates derived from the benthic foraminiferal compilation indicate global temperature decreased by ~1°C from the pre-WOF to WOF time slices (Fig. 14c). An accompanying decrease

of ~40% in CO$_2$ implies a climate sensitivity of ~2.5 (i.e. a 1°C decrease in temperature for a 40% decrease in CO$_2$ equates to a decrease of 2.5°C for a halving of CO$_2$). As noted above, however, our temperature calculations assume ice-free conditions. If cooling was associated with ice growth, a portion of the positive shift in $\delta^{18}$O should be attributed to this increase in ice volume, which would lead to a smaller decrease in temperature and, therefore, lower climate sensitivity.

Our estimates for CO$_2$ in the underlying facies are consistent with published estimates for CO$_2$ in the Paleocene (Fig. 14e), with best fit shown by terrestrial OM and *n*-alkanes CE7. This suggests that CO$_2$ levels during Waipawa deposition were in the range of 200–300 ppm, i.e. below present-day levels and low enough for polar ice sheet growth, at least in the Southern Hemisphere. CE8

**Table 1.** Parameters used to derive atmospheric $CO_2$ estimates for Waipawa and underlying organofacies. TS6

| | Age (Ma) | $\delta^{13}C$[a] | $\delta^{13}C[CO_2]$[b] | $\Delta(\Delta^{13}C)$[c] | $CO_2$[c] | % decrease[d] | Sensitivity[e] |
|---|---|---|---|---|---|---|---|
| **Terrestrial OM** | | | | | | | |
| Pre-PETM ($t = 0$) | 56.5–56.1 | −22.00 | −5.80 | | 385 | | |
| OM-rich WOF | 59–58.5 | −18.00 | −5.00 | −3.33 | 208 | 0.37 | 2.7 |
| Pre-WOF | 60–59.5 | −21.00 | −5.50 | −0.73 | 333 | | |
| **$n$-Alkanes** | | | | | | | |
| Pre-PETM ($t = 0$) | 56.5–56.1 | −26.00 | −5.80 | | 385 | | |
| OM-rich WOF | 59–58.5 | −23.45 | −5.00 | −1.85 | 270 | 0.44 | 2.3 |
| Pre-WOF | 60–59.5 | −26.72 | −5.50 | 1.06 | 485 | | |
| **Fatty acids** | | | | | | | |
| Pre-PETM ($t = 0$) | 56.5–56.1 | −26.00 | −5.80 | | 385 | | |
| OM-rich WOF | 59–58.5 | −25.00 | −5.00 | −0.23 | 368 | 0.40 | 2.5 |
| Pre-WOF | 60-059.5 | −27.60 | −5.50 | 1.99 | 609 | | |

[a] Pre-PETM values from Chen et al. (2014). [b] Calculated from Westerhold et al. (2020) using method of Tipple et al. (2010). [c] Calculated using Eqs. (1) and (2) from Cui and Schubert (2016, 2018). [d] Percentage decrease in $CO_2$ in Waipawa organofacies. [e] Decrease in temperature (°C) with one halving of $CO_2$.

## 5   Discussion

### 5.1   New insights into the depositional setting of Waipawa organofacies

Waipawa organofacies is widely distributed in the southwest Pacific (Fig. 1), occurring in most of New Zealand's sedimentary basins as well as the East Tasman Plateau (Hollis et al., 2014). It is inferred to have been deposited at a range of paleo-depths, from inner shelf to middle slope (Moore, 1988; Schiøler et al., 2010; Naeher et al., 2019), and within a narrow time window of $\leq 1$ Myr in the early late Paleocene ($\sim$ 59 Ma). Previously, Waipawa organofacies was thought to have been deposited during a regression or lowstand following a base level fall (Schiøler et al., 2010; Hollis et al., 2014). However, benthic foraminiferal assemblages in the Taylor White section indicate a general deepening or transgressive trend from the underlying Whangai and into the overlying Wanstead Formation (Naeher et al., 2019). This suggests an alternative interpretation of the palynofacies assemblages within the Waipawa organofacies is required. If we examine how palynofacies assemblages vary in relation to proximity to paleo-shoreline for our studied sections (Fig. 15), we find that a conventional distribution is evident for the underlying facies, i.e. Whangai or Wickliffe formations (Fig. 15a). Terrestrial components (i.e. phytoclasts + terrestrial palynomorphs) tend to decrease, whereas marine elements (amorphous organic matter + marine palynomorphs) increase with water depth and hence with distance from shore. However, the relationship is reversed for Waipawa organofacies: terrestrial components increase, whereas marine components decrease with water depth and distance from shore (Fig. 15b). Degraded phytoclasts, in particular, exhibit a pronounced in-

crease in abundance in the deeper and more distal sections. This distribution of terrestrial OM is supported by biomarkers, which show that terrestrial influence is strongest in sections with the thickest accumulations of Waipawa sediments (e.g. Taylor White and Orui-1A) and weakest in both more proximal (Te Hoe) and most distal (Mead, Ben More) settings (Fig. 16).

We interpret this to indicate that Waipawa organofacies is the result of a rapid influx of terrestrial OM into the marine environment. Corroborating evidence for a marked increase in terrestrial runoff at this time has recently been reported in a study of biomarkers and dinoflagellate paleoecology from ODP Site 1172 (Bijl et al., 2021). CE9 The foraminiferal data (Naeher et al., 2019) suggest that this runoff event occurred while basins were progressively deepening, which is consistent with long-term passive-margin subsidence throughout New Zealand (King et al., 1999). This discovery resolves a long-standing enigma in local geology. In the sedimentary basins to the east of New Zealand, a transition from siliciclastic to hemipelagic sedimentation occurs in the late Paleocene. In the central and northern East Coast Basin, the siliciclastic Whangai facies is overlain by the hemipelagic Wanstead facies (Field and Uruski, 1997). Both units are inferred to have been deposited at bathyal depths. If Waipawa organofacies is present, it occurs at the facies transition. It is difficult to develop a credible depositional model in which a nearshore Waipawa facies was sandwiched between two bathyal units, the overlying one being deeper bathyal than the underlying one.

Although the stratigraphic sections used in this study are primarily on the eastern margin of Paleocene Zealandia, there is evidence from the Taranaki Basin (Fig. 2) that intensified terrestrial runoff affected the entire landmass. Much of the

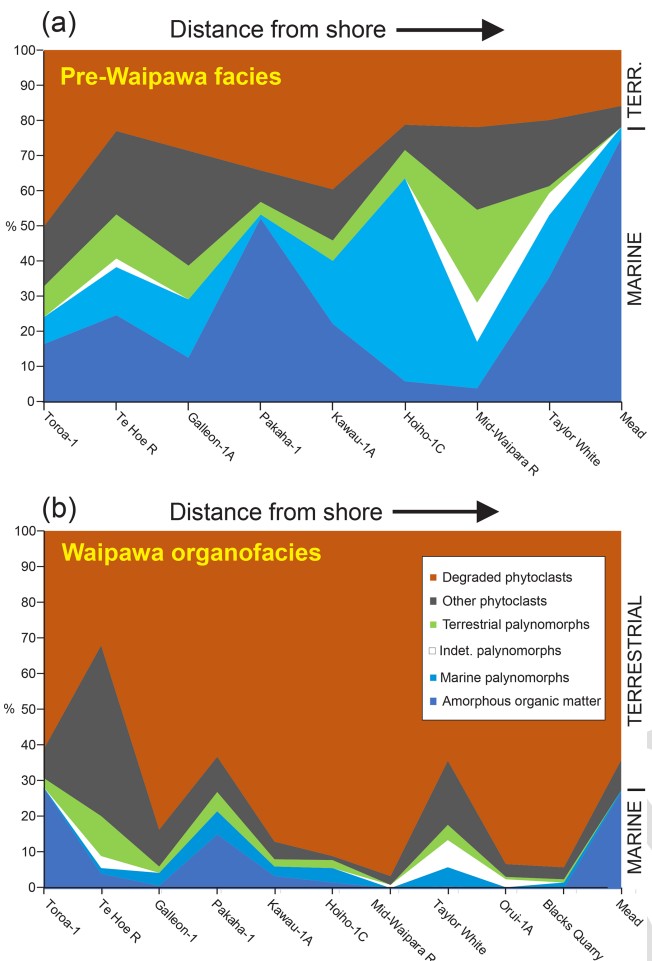

**Figure 15.** Palynofacies variation in relation to inferred relative distance from shore for **(a)** Waipawa organofacies and **(b)** the underlying Whangai or correlative facies, based on mean values. See Table S1 for paleo-depth assessments for each locality.

basin was non-marine to shallow marine in the Paleocene, but in the offshore Reinga sub-basin to the north a 26 m thick interval of Waipawa organofacies is present in the Waka-Nui 1 exploration well (Stagpoole et al., 2009). This suggests

that with further stratigraphic drilling in the offshore western basins, more records of Waipawa organofacies will likely be found.

## 5.2 Global correlations and drivers of Waipawa organofacies

When correlated to deep-sea benthic isotope records (Westerhold et al., 2011, 2020; Littler et al., 2014; Barnet et al., 2019), Waipawa organofacies deposition is found to coincide with a minimum in deep-sea temperatures and the onset of the ∼ 2 Myr long PCIM (Fig. 1). Several factors have

been implicated in the long-term trends in Paleocene temperature and carbon cycling. Cooling from the late early to

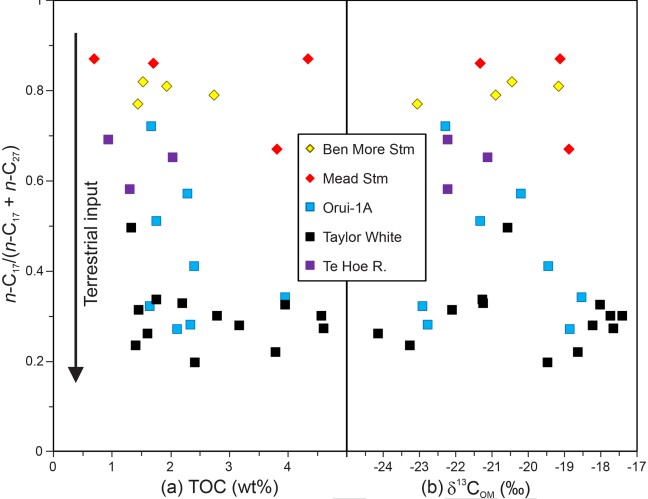

**Figure 16.** Relative abundance of $C_{17}$ $n$-alkanes in relation to **(a)** TOC and **(b)** $\delta^{13}C_{OM}$ as a guide to aquatic/marine input in Waipawa organofacies at five sections representing a middle shelf to middle slope transect: Te Hoe River → Taylor White → Orui-1A → Mead Stream → Ben More Stream.

middle Paleocene (63–60 Ma) followed by warming in the late Paleocene (58–56 Ma) has been linked to global trends in volcanism (Westerhold et al., 2011), carbon cycling (Komar et al., 2013), continental rifting (Brune et al., 2017) and

20 tectonism (Beck et al., 1995; Kurtz et al., 2003; Rotich et al., 2020). Kurtz et al. (2003) argued that the PCIM was caused by the accumulation of terrestrial carbon during tectonic uplift CE10. They point to the vast coal deposits of the Powder River basin, which represent the swamps that replaced North

America's epeiric seas during the Laramide orogeny. Our observation that the OM in the Waipawa organofacies is also terrestrial adds another, albeit offshore, sink for terrestrial organic carbon at this time. Hilting et al. (2008) also interpret the PCIM to be a time of enhanced terrestrial carbon burial

and reduced $CO_2$ levels. Terrestrial carbon burial is modelled to have reduced dissolved inorganic carbon (DIC) in the global ocean (Hilting et al., 2008). This is consistent with our correlation of Waipawa organofacies with an interval of carbonate dissolution during the initial part of the PCIM (Fig. 1).

During the second part of the PCIM, carbonate accumulation recovered at the same time as deep-sea temperatures began to increase (Barnet et al., 2019), suggesting that a new source of carbon offset the effects of carbon burial, such as the $CO_2$ outgassing from the second phase of North Atlantic Igneous

Province (NAIP) volcanism (Westerhold et al., 2011) or oxidation of existing carbon reservoirs (Komar et al., 2013).

Paleocene sediments with organic $\delta^{13}C$ values in the same range (−24‰ to −17‰) as those reported for Waipawa organofacies have been reported from non-marine Paleocene

sections in China (Clyde et al., 2008) and Argentina (Hyland et al., 2015). In the middle-Paleocene Chijiang Basin

section in China, a positive $\delta^{13}C_{OM}$ excursion occurs from a background of values of $-24$‰ to $-22$‰ to $-17$‰ (Clyde et al., 2008). This demonstrates that similar processes may have affected terrestrial plant matter in regions beyond the southwest Pacific, i.e. $CO_2$-controlled carbon fractionation with or without lignin degradation. However, magnetostratigraphy for this section indicates that the excursion predates the Waipawa event by $\sim 1$ Myr. In the Salto Basin section in Argentina, a middle–late Paleocene section comprises two cycles in which $\delta^{13}C_{OM}$ ranges from depleted values of $-26$‰ to $-25$‰ to enriched values of $-21$‰, again a shift similar to that recorded by mid-Waipara HMW fatty acids CE11. The enriched values are linked to proxies for lower temperature and reduced precipitation (Hyland et al., 2015). A 30 m thick interval with enriched $\delta^{13}C_{OM}$ values at the base of the late Paleocene (lower Chron 26n) is very sparsely sampled and may correlate with the Waipawa event. It is sandwiched between two intervals with more depleted $\delta^{13}C_{OM}$ values that are correlated with a two-phase ELPE but may represent an ELPE and "*" events.

Some have argued that the ELPE is a hyperthermal (Bernaola et al., 2007), but expression of the negative CIE is variable and evidence for warming is equivocal. Possible evidence is seen at ODP Site 1262 (Littler et al., 2014; Barnet et al., 2019), where individual samples record light $\delta^{18}O$ values. However, these samples are anomalies against a background of relatively heavy $\delta^{18}O$ values and may be more plausibly explained by downslope transport of individual benthic foraminifera. Moreover, a burrowed horizon near the base of the ELPE at Site 1262 (1262B-18H-4, 97–127 cm) suggests the presence of an unconformity and, therefore, an incomplete record.

The high-resolution records from sites 1209 and 1262 (Fig. 1) indicate that the ELPE marks a significant turning point in Paleocene climate and carbon cycling: the termination of a long-term (4 Myr) cooling trend followed by a prolonged period of carbon burial, i.e. the PCIM. Significantly, benthic foraminiferal $\delta^{18}O$ and $\delta^{13}C$ trends are coupled from 63 to 59 Ma, with positive shifts in both parameters suggesting that cooling was associated with declining atmospheric $CO_2$ (Fig. 1). From 59 to 58 Ma, the trends are not coupled: $\delta^{13}C$ continues to increase while $\delta^{18}O$ either decreases or remains stable. From 58 Ma to at least the early Eocene, the records are once more coupled. This interval of uncoupled isotope records begins with the ELPE and ends with the "*" event. Barnet et al. (2019) show that the interval also contrasts with intervals below and above by having much less coherent eccentricity phasing, suggesting the influence of non-orbital climate drivers, such as the tectonic events noted above. Carbon cycle modelling also indicates that this interval is followed by a shift to a net decrease in organic carbon burial, with oxidation of carbon reservoirs driving the subsequent warming trend that culminates in the early Eocene climatic optimum (Komar et al., 2013). This shift may be linked to changes in deep-water circulation. Neodymium iso-

topes point to intensified deep-water exchange between the North and South Atlantic, probably due to the deepening of the central Atlantic Rio Grande rise (Batenburg et al., 2018).

In summary, there is wide-ranging evidence that the interval of Waipawa organofacies deposition is linked to a significant turning point in Paleogene climate and carbon cycling, transitioning from the cooler climate conditions and relatively high rates of organic carbon burial of the middle Paleocene to the warming climate and lower rates of organic carbon burial of the late Paleocene and early Eocene.

## 6   Conclusions

Correlation of Paleocene sedimentary successions in the southwest Pacific with deep-sea stable isotope records has revealed that the deposition of organic-rich sedimentary facies on the continental shelf and slope of New Zealand and eastern Australia, called Waipawa organofacies, occurred over a period of $\sim 700$ kyr within an episode of global cooling and increased carbon burial between 60 and 58 Ma (Figs. 1, 13). The sequence of events that led to the deposition of the Waipawa organofacies is highly unusual, if not unique, in the geological record. The organic-rich nature of the marine mud facies is due mainly to massive input of degraded woody plant matter. Both this plant matter and a subordinate amount of marine algal material are collectively enriched in $^{13}C$ by $\sim 7$‰–10‰. The CIE is thought to be amplified by degradation processes during transport and deposition of terrestrial OM. However, a residual CIE of $\sim 3$‰ in terrestrial biomarkers is inferred to represent a 40 % reduction in atmospheric $CO_2$ levels. Episodes of $CO_2$ drawdown and climatic cooling are common in the geological record, but this event appears unique in resulting in the regionally widespread and rapid deposition of degraded terrestrial plant matter. We postulate a scenario in which four independent processes are at play.

i. *A pause in North Atlantic volcanism.* Climate cooled in the middle Paleocene as the first phase of NAIP volcanism subsided and $CO_2$ emissions decreased. Climatic cooling is likely to have increased the storage of carbon as biogenic methane in continental shelves (Dickens, 2003) and in high-latitude peatlands and permafrost (DeConto et al., 2012). This would have led to a positive feedback in which carbon burial caused further lowering of atmospheric $CO_2$ and further cooling. Climate then warmed through the later Paleocene and Eocene as the second phase of NAIP volcanism ramped up.

ii. *North American tectonism.* Between these two volcanic $CO_2$-modulated climate shifts, the Laramide uplift event is thought to have turned the vast epeiric seas in North America into peat swamps, forming a large carbon sink (Kurtz et al., 2003) and leading to further $CO_2$ drawdown, cooling and carbonate dissolution in

the deep sea (Hilting et al., 2008). A significant fall in sea level is inferred to have occurred at around this time, namely the Th2 event of Hardenbol et al. (1998) and the Pa2b event of Kominz et al. (2008). Harris et al. (2010) argue that this event corresponds to a glacioeustatic fall in sea level of $\sim 15$ m. This fall in sea level has also been linked to a large system of fluid escape pipes discovered in late Paleocene sediments offshore eastern New Zealand (Bertoni et al., 2019).

iii. *Southwest Pacific tectonism.* In the context of long-term passive margin subsidence and the opening of the Tasman Sea, rapid basinal deepening occurred through much of the southwest Pacific in the late Paleocene as evidenced by the transition from siliciclastic to hemipelagic to pelagic carbonate facies in the Great South, Canterbury, East Coast and North Slope basins east of New Zealand (Field and Browne, 1989; Field and Uruski, 1997; Cook et al., 1999; Isaac et al., 1994; King et al., 1999). Foraminiferal assemblages in the Taylor White succession confirm that this transgression progresses through the Waipawa Formation and into the overlying Wanstead Formation (Naeher et al., CE12 A similar geological history is inferred for the Tasmanian margin where a major runoff event is linked to Waipawa deposition (Hill and Exon, 2004; Bijl et al., 2021). Therefore, the only plausible explanation for a rapid influx of terrestrial plant matter is one or more eustatic falls in sea level, eroding coastal vegetation and flushing the debris into offshore basins.

iv. *Orbital forcing.* The Paleocene includes five 2.4 Myr eccentricity cycles, each of which comprise six 405 kyr cycles (Barnet et al., 2019). Waipawa organofacies deposition occurred during a $\sim 1$ Myr decline in eccentricity forcing with the maximum coinciding with the ELPE and the minimum coinciding with the "*" event (Fig. 13). Three of the other eccentricity minima occurred at times when the NAIP was active, whereas the first occurred at $\sim 64.5$ Ma and corresponds with a pronounced regional cooling event (Taylor et al., 2018).

All four processes appear to have had a role in creating Waipawa organofacies. A relatively warm climate from the late early to middle Paleocene may have led to the expansion of terrestrial vegetation. Cessation of CO₂ emissions from NAIP volcanism may have coincided with an eccentricity minimum to cause rapid cooling. Lower CO₂ levels may also be attributed to increased carbon burial in North America swamps coupled with sequestration of biogenic methane in continental shelves and high-latitude permafrost as positive feedbacks. Cooling is inferred to have led to the growth of ephemeral ice sheets, which caused sea levels to fall and the erosion of well-vegetated coastal areas in Zealandia and parts of Australia, and the accumulation of abundant terrestrial plant matter in marine basins. It is possible that the interplay of eccentricity-modulated climate cycles and basinal subsidence led to pulses of erosion, deposition and redeposition, which provides a mechanism for seafloor degradation of woody plant matter to occur prior to remobilization and redeposition. A similar scenario during late Miocene lowstands led to large accumulations of terrestrial OM in the deep-water Kutei Basin in East Kalimantan (Saller et al., 2006). Admittedly, the importance of orbital forcing is uncertain because of the low level of coherency in eccentricity phasing from 59.5 to 58 Ma (Barnet et al., 2019).

It now seems likely that the variation in TOC and $\delta^{13}C_{OM}$ seen in expanded records of Waipawa organofacies such as the Taylor White section can be correlated to similar scales of variation in the high-resolution benthic isotope records. Confirmation of this would require more closely spaced sampling than has been possible in this study, ideally as part of a stratigraphic drilling project.

Previous studies of past greenhouse climates of the early Paleogene have identified short-lived global warming events, termed hyperthermals, that have been the subject of numerous studies because of the insights they offer for understanding projected global warming scenarios (Zachos et al., 2008; Sexton et al., 2011; Westerhold et al., 2018; Barnet et al., 2019). Within the same time interval, we have identified a similarly short-lived cooling event, which we term a hypothermal, that has potential to offer insights into how the planet may recover from global warming.

**Data availability.** All data generated in this study are tabulated in the Supplement.

**Supplement.** The supplement related to this article is available online at: https://doi.org/10.5194/cp-18-1-2022-supplement.

**Author contributions.** CJH, SN, CDC and RS designed, directed and led the study. SN, GTV, CDC, JD, XL, BDAN and KWRT acquired and analysed the data. CJH, SN, CDC, BDAN, RDP and RS interpreted the data with input from all other authors. CJH, SN, RS, BDAN and RDP wrote the paper with contributions from all co-authors.

**Competing interests.** The contact author has declared that neither they nor their co-authors have any competing interests.

**Disclaimer.** Publisher's note: Copernicus Publications remains neutral with regard to jurisdictional claims in published maps and institutional affiliations.

**Acknowledgements.** We thank Randall McDonnell, Sonja Bermudez and Henry Gard for crushing and powdering rock sam-

ples, Roger Tremain for preparing samples for palynological analyses, and Andy Phillips, Jannine Cooper and Joan Fitzgerald for organic-matter carbon isotope analyses. Applied Petroleum Technology (Norway) provided lipid biomarker analyses and the International Ocean Discovery Programme provided samples from ODP Site 1172. We thank Poul Schiøler, Michael Tayler, Kate Littler, James Barnet, Thomas Westerhold, Jerry Dickens and Matt Huber for sharing data and providing valuable insights and stimulating discussion in the course of this work. Internal institutional reviews by Dominic Strogen and Erica Crouch and a community comment by Steve Killops as well as reviews by Peter Bijl and an anonymous referee during the peer review process greatly improved the paper CE13.

**Financial support.** Primary funding for this study came from the Ministry of Business, Innovation and Employment (MBIE), New Zealand, as part of the GNS Science-led programme "Understanding petroleum source rocks, fluids, and plumbing systems in New Zealand basins: a critical basis for future oil and gas discoveries" (Contract C05X1507). Initial work was supported by the New Zealand Marsden Fund (Contract GNS0702). Work conducted at the University of Bristol was supported by the NERC through partial funding of the National Environmental Isotope Facility (NEIF; contract no. NE/V003917/1), the European Research Council under the European Union's Seventh Framework Programme (FP/2007-2013) and European Research Council (grant no. 340923) for funding GC-MS capabilities, the NERC (contract no. NE/V003917/1) and the University of Bristol for funding the GC-IRMS capabilities, and funding for B. David A. Naafs TS7 from a Royal Society Tata University Research Fellowship.

**Review statement.** This paper was edited by Appy Sluijs and reviewed by Peter Bijl and one anonymous referee.

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

## Remarks from the language copy-editor

## Remarks from the typesetter

**TS2** Please note that only technical changes can be made at this stage of the publication process. According to our standards, changes to the content of the paper, such as this one, must first be approved by the editor, as data have already been reviewed, discussed and approved. Please provide a detailed explanation for each change in values (i.e. please highlight each instance and explain what must be changed and why) that can be forwarded to the editor. The same applies to the new figure and table. Please note that this process will be available for readers to see online after publication. Upon approval, we will make the appropriate changes. Thank you for your understanding.