# Peer review of "Late Paleocene CO2 drawdown, climatic cooling, and terrestrial denudation in the southwest Pacific"

_Climate of the Past, 2021_

## Author Comment (AC1)

**Comments from Steve Killops**

*At the request of Chris, here are some thoughts I hope will be helpful*

The authors greatly appreciate these comments from Dr Killops and respond to each point below.

*Intro – it seems difficult to tie methane hydrate formation to the interval if it's associated with a sea-level fall, given the interplay of temperature and pressure. The effects of global cooling are counteracted by eustatic sea-level fall, so estimating whether methane hydrate can account for the C isotopic changes is tricky.*

We made an error in referring to continental shelves, where the effects of a fall in sea level might be expected, but hydrates typically form in deeper water >1000 m, where these effects would be minimal. We will change "on continental shelves" to "at continental margins", which is in line with the model presented by Dickens (2003). No need for further changes as this a minor element in the paper.

*Fig 3 – the positive correlation might be easier to see with a linear TOC axis. The fits aren't impressive from the R2 values. I wonder if linear correlations are really important, as it's likely that CO2 levels would have to fall quite low (due to significant local, if not global, draw-down) before fractionation is affected, which could mean that although TOC and d13C are correlated, the relationship might not be linear.*

The primary purpose of this figure is to emphasise a feature that is central to the definition of Waipawa organofacies: that TOC and bulk organic $\delta^{13}C$ are correlated. The various reasons for this correlation, including $CO_2$, are addressed subsequently. A log scale is chosen to make it easier to view trends within sections with varying ranges in TOC (e.g., Glendhu vs Black's Quarry).

*Lines 245+ – Could the fluorescence characteristics of the amorphous OM help distinguish algal from higher plant contributions, if available? The moderate linear regression correlations are not convincing – particularly when the Whangai and Wanstead samples are removed from Fig 5. Fig 6 is more convincing.*

Fluorescence was used to differentiate algal material from phytoclasts. We agree that the correlations in Fig. 5 are not that strong, but the main point of this figure is to show how palynofacies fractions vary between organofacies. The bulk and light fractions exhibit a trend in which $\delta^{13}C$ increases as the proportion of degraded phytoclasts increases (Whangai/Wanstead to OM-poor to OM-rich Waipawa facies), whereas for the heavy fraction there is much less variation in $\delta^{13}C$, suggesting less mixing of sources and that the 5 per mil offset in $\delta^{13}C$ between the Whangai and the degraded phytoclasts-dominated Waipawa may be mainly due to the process of degradation.

We further note that, although the linear correlations in Fig. 5 are not strong, they are, nonetheless, statistically significant. Recognition of degraded phytoclasts vs other phytoclasts is to some degree subjective, based on visual identification and commonly requiring a judgement call on what may be slightly degraded versus non-degraded. The two parameters ($\delta^{13}C_{OM}$ and % degraded phytoclasts) are also based on different fractions of the entire OM assemblage. For palynofacies, material <6 micron in size is filtered out because it is generally too small for reliable visual identification of palynofacies classes. The fact that moderate correlations exist between $\delta^{13}C_{OM}$ and % degraded phytoclasts despite these limitations, gives confidence that the relationships are indeed intrinsic characteristics of the OM assemblages.

*5.3.1 – This paragraph seems a little problematical. If phytoclasts are significantly degraded, carbohydrate residues will be almost non-existent (as noted in 5.3.2). Such a low linear regression coefficient might be considered to rule out correlation. Does d34Sorg say anything about sulphate supply and likely S incorporation?*

Good point. We will add an additional comment that phytoclast degradation also impacts the preservation of carbohydrates, supporting our conclusion that sulfurization cannot explain the $^{13}$C enrichment. We discussed the relationship between bulk $δ^{34}$S and redox conditions in Naeher et al. (2019).

*Samples labelled TW-15 and TW-17 do not appear to correspond in Fig 7 – assuming these outliers in (b) are correctly labelled, the problem is with (a).*

Thanks for noting this error in labelling. We will remove all labels from this figure because, as the reviewer notes, the correlation is weak and there is no need to interrogate the data in further detail.

*There's an assumption about the origin and abundance of naphthalene in interpreting Fig 7a that would be worth stating so the reader knows why the ratio works in the way proposed.*

We will add this statement to the caption of Fig. 7. "Naphthalene is used to normalise these compounds because it is a generic compound independent of source".

*5.3.2 – Fitting a linear trend to the data in Fig.8b seems a bit optimistic. The figure legend is a bit confusing as it suggests the difference between low and high TOC samples is being emphasised, but that requires examining the TOC values by each data point. How about a different symbol shape for each TOC group to make it stand out better? The Sofer distinction between terrestrial and marine is contentious and was based on oil data, rather than immature sediment extracts, so the CV value interpretation is a bit shakey. (a) is a more useful plot in terms of variation in d13C with TOC, so it could be worth considering omitting (b).*

Fig. 8b was an effort to emphasise the terrestrial nature of the OM but we agree that is problematic applying a method designed for oils to sediment extracts. As the terrestrial nature of the OM has already been demonstrated, this figure is not required. As suggested, we will redraft Fig. 8a (now Fig. 8 – see below) to distinguish the organofacies (TOC groups).

*Some discussion would be helpful of why d13C sat is not affected by degradation when the dominantly lignin derived aromatic value is. Is the inference that epicuticular waxes are preferentially preserved, so the lighter d13C of the higher plant n-alkanes cf phytoplankton biomass is conserved?*

We will clarify that there is a significant positive correlation between terrestrial OM derived from palynofacies and the abundance of aromatics relative the saturated fraction, and especially so with degraded phytoclasts ($r^2$ = 0.54, n = 20, Taylor White section). Although higher plant n-alkanes are abundant in Waipawa organofacies, the total saturated fraction represents a mixture of terrestrial and marine OM and the latter will not have been affected by transport-related degradation. Indeed, the difference between aromatic and saturated $δ^{13}$C may provide a further clue to the component of the $δ^{13}$C excursion that can be linked to degradation. Palynofacies study indicates that there is very little leaf cuticle present.

*line 337 – it might be better to say that one explanation for the position of TW-19 in Fig 8a is that it contains more marine OM than suggested by palynofacies results. The present wording looks a little like adjusting the results to fit the model.*

We will completely revise and simplify this section so that it is restricted to considering the differences between aromatic and saturated fractions.

*line 340-1 – a ref to reducing conditions in NZ peats would be good.*

With simplification of this section, this reference is no longer needed.

*Final paragraph notes the varying marine OM contribution, but is it worth discussing whether differing terrestrial contributions, reworking and transport to the depositional environment could be a major cause of the observed variation in bulk d13C values?*

We will revise concluding remarks for this section accordingly.

*5.4 lines 380-4 – As noted, the C-number range is usually a reasonable proxy for terrestrial vs aquatic primary production. However, the dominance of Sarcinochrysidales suggests that we may not be dealing with the usual marine primary producers. It's worth bearing in mind that algae such as Botryococcus produces long-chain n-alkanes (and C29 steroids).*

C-number range is a reasonable proxy for terrestrial vs aquatic primary production and is used in many published studies. It appears to work OK in this study and is consistent with several other such proxies that we have used, i.e., we have not relied solely on C-number range.

We find no evidence of *Botryococcus* in Waipawa sediments. Botryococcane is common in NZ lacustrine sediments but has not been found in Waipawa samples. While it's true that the dominance of C30 steranes points to an abundance of unusual algae, there is no evidence that these algae had an unusual C-number range. Therefore, we make no change to the text.

*6.2 lines 446-7 – Evidence of fungal degradation of lignin might be sought from perylene. Monitoring m/z 252 in aromatics fractions gives both perylene and benzopyrenes (pyrolytic PAHs), so you can combine looking at lignin degradation with the influence of wildfires (which might show some negative correlation with cooling).*

This is an interesting topic for future study. Preliminary data do, in fact, suggest that perylene is abundant in Waipawa organofacies; however, there is no correlation with $\delta^{13}$C. Therefore, the possible link between fungal degradation and $^{13}$C enrichment cannot be demonstrated with this proxy. Pyrene is correlated with $\delta^{13}$C, suggesting a possible link with wildfires but further analyses would be needed to pursue this further.

*lines 479-84. As relative abundances are being assessed, could suppression of marine primary production help overcome the problem of deepening but relatively more terrestrial contribution?*

The magnitude of the increase in terrestrial OM in the Waipawa Formation indicates that it reflects a massive influx of terrestrial plant matter, not simply a relative increase due to decreased marine OM input. Many indicators suggest that marine primary production increased during Waipawa deposition (Hollis et al., 2014; Hines et al., 2019; Naeher et al., 2019).

*The prominence of 24-n-propylcholesteroid producing alga seems unique to the Waipawa Fm and suggests there is something funny going on. If these C30s often dominate steranes in Waipawa samples, could it suggest that the large terrestrial OM input is pretty heavily reworked (with steroid removal)?*

Whilst it is true that the 24-n-propylcholestanes are of exceptionally high abundance within the Waipawa Fm, we attribute this to the particular water column chemistry conditions resulting from the massive influx of terrestrial OM, rather than to heavy reworking of the terrestrial OM

component. Our palynofacies analyses do not provide any clear evidence for such heavy reworking beyond simple transportation and deposition of the terrestrial OM. The terrestrial OM is dominated by brown phytoclasts, not opaque (black), highly oxidised OM, which would have been expected if the OM had been heavily reworked.

*Kerogen d13C is likely to be more useful than total organic extract or fraction d13C when assessing sources of the bulk of OM, but the method suggests only extract measurement or CSIA was undertaken. From Fig 1 and related text it looks like kerogen d13C was obtained, so some clarification in the methods and a comment in the text about what d13Com represents would be helpful.*

We did not include any kerogen fraction $\delta^{13}$C measurements within this study. Rather we used $\delta^{13}$C analyses of decalcified bulk samples (method as described in Naeher et al. 2019) comprising both the kerogen and bitumen fractions. The use of decalcified rock samples for bulk carbon content and isotope analyses has now been made clearer in the revised text.

*The CSIA data in Fig 12 are very spikey, which often happens if isolation of n-alkanes has not worked too well. It's useful to check recovery by GC. APT has been unable to reproduce the Grice et al (2008) method, which tends to give poor recovery and very spikey data. APT has developed a reliable urea adduction method now which gives good n-alkane recovery and smooth d13C trends. In Fig 12 the deviation between the two groups at nC27+ looks dependable, but it would be dangerous to go further than that.*

The deviation between the two groups are all we are aiming to show for this figure.

*As pointed out in the m/s, the big problem is what the background d13C signatures may be during Waipawa deposition for the end-member terrestrial and aquatic OM contributions – in order then to estimate relative amounts of each contribution. One method that might be useful to examine terrestrial vs aquatic are plots from the pyrolysis-GC data that APT produced for GNS on many of the study samples (assuming no ownership issues). There are three ternary plots in the attached pdf that may be helpful. It might be possible to look at combinations of parameters from this data along with d13Ckero via multivariate stats to derive estimates of the terrestrial-aquatic balance in each sample, rather than using end-member d13C values for the Whangai and Wanstead, which may not be representative. Possibly a long shot, but who knows? If there is a lot of inertinite in the mix, that could really drag the d13C down but not affect TOC so much – the final ternary might help assess that.*

As noted above, we did not analyse $\delta^{13}$C specifically of the kerogen fraction, thus precluding comparison between $\delta^{13}$C kero and pyrolysis-GC-derived compounds as suggested by Dr Killops. We would also note that the ternary plot templates suggested by Dr Killops to investigate the relative proportions of terrestrial and marine OM are very generic in nature and based on international data sets, whereas our study has quantified the proportions of terrestrial and marine organic matter more directly using palynofacies analyses. However, it is still not clear what analytical approach might best be used to ascertain the respective shifts in $\delta^{13}$C of the separate terrestrial and marine organic matter assemblages from the underlying facies to the Waipawa. For now, we have estimated the $\delta^{13}$C shift for the bulk OM assemblages by comparison between the Waipawa and underlying facies.

Dr Killops also speculates whether high inertinite contents within the Waipawa Fm might have affected the relationship between $\delta^{13}$C and TOC. However, inertinite contents are not high within the Waipawa Formation.

Additional References

Dickens, G.R. (2003) Rethinking the global carbon cycle with a large, dynamic and microbially mediated gas hydrate capacitor. Earth and Planetary Science Letters 213, 169-183.

---

## Author Comment (AC2)

**Review by Peter Bijl**

*The authors present a really compelling dataset representing local depositional setting and terrestrial climate of the Paleocene of New Zealand. The geographical spread of the records, around the north and east coast of the continent, makes it a comprehensive and complete overview, with compelling implications. I have some comments on the way the study is introduced and discussed, but these should be easily fixable, either as reply or in a revised draft.*

We appreciate the positive appraisal of this study and the thoughtful comments below.

*I understand the introduction gives the potential importance of understanding Waipawa organofacies deposition in the context of past climate change, with CO2 drawdown as mechanism and that would fit well with the scope of the journal. However, given the primary focus of the study, to characterize the black shale OM content, and understand the enigmatic enrichment in 13carbon, I would suggest the authors focus the introduction a bit more on existing investigations in other black shales. As it is now, the reader expects a "CO2 drawdown paper" but gets quite detailed analyses of OM composition and geochemistry instead. Meanwhile, the quantification of CO2 drawdown and a convincing argumentation for why the found signals can only be caused by CO2 drawdown, is largely missing. Assessing the way the aims of the paper are introduced may be a bit outside my tasks as a reviewer, but I feel the way it is now has the introduction somewhat disconnected to the bulk of the paper.*

We disagree with the reviewer's view on the introduction. We feel the last paragraph (lines 73–83) explains why we focus on identifying the source of the d13C excursion.

*The aim of the study is to find the cause of the 13C-enriched OM. The authors argue for CO2 drawdown as a cause, and indeed that could be one of the reasons (although there are some others as well). However, the authors add cooling as supporting argument for that (it is cooling, so there must have been a CO2 decline), and I think this drives down a dangerous road towards circular reasoning. First of all, they drive away from all the possible other reasons other than CO2 drawdown of why this region cools. Evidence of Paleocene cold conditions mostly comes from southwest Pacific SST data, which represent at best local signals. The authors mention another reason for regional cooling themselves: increased upwelling of deep water. Benthic foram records might be biased by an unknown amount of ephemeral ice volume, and cannot be taken as paleotemperature proxy as such. Secondly, if the cooling is indeed global, the relation to radiative forcing has the issue that long-term trends in benthic foram d13C (representing carbon cycle) and d18O (representing temperature/ice volume) are out of phase by 1.5 Myrs. Westerhold et al., 2011 provides dissolution as a potential but uncertain reason for this, but as long as this is unresolved, the community has to entertain the idea that this represents a genuine signal, with understanding of the 1.5myr delay unexplained. Then, If the abstract and the rest of the paper reads as if it was shown that CO2 drawdown caused the d13C enrichment, people will use the paper as evidence for CO2 decline in the Paleocene, while actually that conclusion was drawn with the use of (local) SST decline as argument. Then CO2 reconstructions and temperature reconstructions have lost their independence, which is a tricky road.*

We accept that there is a danger of some circular reasoning, but we feel that we have made considerable efforts to explore alternative explanations for $^{13}$C enrichment. There is no question that interest in the Waipawa organofacies has centred on the potential link between the regional cooling reported by Hollis et al. (2014) and the nature of the organofacies – i.e. enrichment in TOC and in $^{13}$C. Positive excursions in δ$^{13}$C are widely associated with $CO_2$ drawdown events, so it makes sense for the initial hypothesis for this study to be: $^{13}$C enrichment in Waipawa organofacies is linked to a

global drawdown in $CO_2$ and global cooling. Much of the study is devoted to testing this hypothesis, searching for alternative explanations, and eventually concluding that not all but some of the enrichment is reasonably explained by a global perturbation to the global carbon cycle (e.g., lines 403–410).

*Other factors may explain why d13C of higher plants might be shifting carbon isotope values over these time scales: lapse rates, for instance (Körner et al., 1988; doi: 10.1007/BF00380063). Could the authors find evidence to exclude the possibility that a change in altitude of the catchment caused some of the d13C excursion in the terrestrial components? I feel that the authors should more carefully exclude other arguments to explain the changes in d13C before the conclusion is drawn that CO2 drawdown caused it. This means acknowledging other potential factors.*

We feel that we have covered the various options for OM sources adequately. The study by Korner et al. (1988) compares lowland to plants at >2500 m altitude. The contribution of vegetation from that altitude to the terrestrial carbon pool would be negligible.

*Another (in my mind) obvious omission in the paper is the implications of the reconstructed intense river runoff signal in the records for local paleogeography and paleoenvironments. Many records of the Waipawa organofacies come from the east coast of NZ, which today, owing to a high mountain range and prevailing westerly winds, is in an intense rain shadow. The observation of intense river runoff in the Paleocene on the east coast of NZ could mean 2 things: (1) prevailing easterly winds in the Paleocene, which is unlikely, but could be verified in model simulations (2) absence of a rain shadow, which means absence of a strong mountain divide. I believe this must be discussed in the paper, and because the evidence for intense runoff is way clearer than the link to atmospheric CO2 drawdown, I would suggest the authors focus their paper towards the implications for local paleogeography, hydrology and paleoenvironment.*

Yes, we agree, that a little more should be said about the implications for hydrology. However, we don't believe the scenario requires a major change in hydrology from present conditions. While the rain shadow is intense in the Southern Alps of the South Island, the prevailing westerly weather system delivers high rainfall to both coasts through drainage systems that drain off the axial ranges to the west and east. The much-studied Waipoua catchment that drains into the Pacific from central North Island carries an extremely high sediment load (East Cape in Fig. 10 of Hicks et al. 2011. Suspended Sediment Yields from New Zealand Rivers. Journal of Hydrology (New Zealand), 50(1), 81–142.). Hydrology alone cannot explain the $^{13}$C enrichment identified in both marine and terrestrial OM. Therefore, we don't think it warrants a change in focus for the article.

Comments in chronological order

*Abstract line 25–27: Authors should be specific about trends vs peak values (cooling versus cold). The 1.5 million year offset means that it is crucial that the authors place the timing of deposition of the Waipawa organofacies and the SST trends relative to the carbon isotope maximum and the oxygen isotope maximum. To me, "cooling" refers to a decreasing trend in temperature, rather than a temperature minimum. Does the deposition of the Waipawa organofacies now coincide best with the benthic foram d13C trends, the d18O maximum or with the SST minimum? Some careful rewording might be needed here to make it really clear.*

Yes, we agree that the wording can be improved. How about this?

Refined age control for Waipawa organofacies indicates that deposition occurred between 59.2 and 58.4 Ma, which coincides an interval of carbonate dissolution in the deep sea that is associated with a Paleocene oxygen isotope maximum (POIM, 59.7–58.1 Ma) and the onset of the Paleocene carbon isotope maximum (PCIM, 59.3–57.4 Ma). This association suggests that Waipawa deposition occurred during a time of cool climatic conditions and increased carbon burial.

*Line 109: the SST data of ODP Site 1172 are indeed published by Hollis et al., 2014, but note that these were updated in Bijl et al., 2021 with higher resolution, and beyond TEX86. Moreover, the primary source for the organic d13C data is Röhl et al., (2004; Geophysical monograph series 151). This should be acknowledged.*

Yes, of course, these references will be added.

*What I would suggest, is that the authors add a small plate presenting wide brightfield microscope images of the palynofacies, highlighting the main palyno groups. It might sound obvious for people working with palynofacies, but given the importance of this dataset for the story, I feel some visual support is warranted.*

Good point. There is a plate in the supplementary files of Naeher et al. (2010, https://ars.els-cdn.com/content/image/1-s2.0-S0264817219301345-mmc14.pdf). However, we can improve on this and will add a plate to this paper, probably also as a supplementary file.

*Line 435–439: See Komar et al., 2013 https://doi.org/10.1002/palo.20060, attempting to reconcile the long-term trends in carbon and oxygen isotopes, and lysocline evolution using a carbon cycle box model.*

Thanks for reminding us of this important paper. Although it mainly focuses on isotope trends after 58 Ma, it helps to emphasise the connection between carbon burial, d13C and the CCD. We will add this to the discussion.

*The inshore-to-offshore trends in Fig. 14 are all but compelling. There is also a good reason why: global average sea level was really low, which means accommodation space was reduced. If the transect does not include sites off the slope (and Mead Stream is top slope, if I am correct), you will not find much of a transect when terrestrial input is so intense.*

Clearly, we need to clarify the paleodepths of the sections and emphasise the evidence from benthic forams. We touch on this in lines 460–470, but need to make it clearer. Most of the sites are upper to mid slope and deepening as NZ basins subsided during passive margin thermal relaxation. So, certainly, base level fall is seen as the likely cause of nearshore erosion, fluvial down-cutting and delivery of terrestrial debris into the slope environment. But the sections are too deep for the increase in terrestrial OM to be solely due to shoaling of ~10-20 m at most.

*In section 6.4, I am losing the connection to the new results. I propose the authors revisit this section to see how it can be more closely connected to their results and implications.*

Yes, we agree. The section lacks a summary that connects the strands of evidence for Waipawa organofacies being a local response to global changes. We'll work on that in the revised MS.

---

## Author Comment (AC3)

**Review by Referee 2**

Hollis et al report new stable carbon isotope measurements of organic matter from sediments deposited on the continental shelf and slope of New Zealand and eastern Australia during the late Paleocene (termed the Waipawa organofacies). The authors identify unusually high  $\delta^{13}C$  values measured within the Waipawa organofacies, consistent with measurements made by others on contemporaneous sections in China and Argentina. The authors use a detailed suite of geochemical analyses (including bulk and compound specific stable isotope analysis) to claim the unusually high  $\delta^{13}C$  values are caused by a combination of lignin degradation and low  $CO_2$  levels. Associated with this event is global cooling (and growth of ice sheets and fall in sea level) that likely resulted from lower atmospheric  $CO_2$  (evidenced by the high  $\delta^{13}C$  values), which may have been caused by reduced volcanism and increased carbon burial.

The authors make the connection between the high  $\delta^{13}C$  values and low  $CO_2$ , but a quantitative estimate of  $CO_2$  is lacking. Using the terrestrial  $\delta^{13}C$  data to quantify  $CO_2$  would allow for a more useful comparison of  $CO_2$  and temperature, and greatly improve what is presently a very qualitative comparison (high  $\delta^{13}C = low CO_2$  and cooling = low  $CO_2$ ). This is particularly important given that "The relationship between temperature and atmospheric greenhouse gas levels through the Paleocene is very poorly resolved..." (40) and the authors state (75), "we explore the possibility that this 13C enrichment of bulk OM reflects a short-lived drawdown in atmospheric CO2, reflecting the relationship in carbon isotope discrimination between atmospheric CO2 and C3 plant biomass (Cui and Schubert, 2016, 2017, 2018; Schubert and Jahren, 2012, 2018)." Yet, any determination of  $CO_2$  using this relationship is conspicuously absent.

Furthermore, the authors later state (290), "Only by accounting for potential processes of 13Cenrichment during OM transportation, deposition and early diagenesis it is possible to identify any residual enrichment that may be related to a drawdown in atmospheric CO2 levels." Why do all this if  $CO_2$  is not going to be estimated quantitatively (even if only a back of the envelope calculation to show a possible range of  $CO_2$  drawdowns, given possible marine influences, and autogenic processes)?

Alternatively, the authors could calculate  $CO_2$  given their interpretation that (376-377), "the pristane CIE implies that the primary terrestrial substrate is enriched in 13C by ~4‰." The authors could also calculate  $CO_2$  for a range of CIE magnitudes, to show the magnitude of  $CO_2$  change that would be required to get any size CIE. It would certainly help to better answer the question of whether a drawdown in  $CO_2$  is a plausible explanation for the  $\delta^{13}C$  trends and the observed cooling (the current assumption is there was cooling therefore  $CO_2$  must have decreased). Is the purported 20-30% decrease in  $CO_2$  required for a 1 °C decrease in deep sea temperature (455) consistent with  $CO_2$  estimated assuming a +4‰ terrestrial CIE (based on the terrestrial CIE)? If so, that would greatly support the stated conclusions linking high  $\delta^{13}C$  to low  $CO_2$  (and the various processes indicated within). If not, it may suggest climate sensitivity differed from the 3 °C assumed here, which would also be an interesting result. Much of the work to assess climate sensitivity in the Paleogene has focused on the warmest periods.

Besides, the aquatic sources show a similar 2-4‰ shift to the terrestrial sources (380-382). If so, why does the relative terrestrial vs aquatic influences matter? Both show similar magnitude CIE, so why would the % terrestrial affect determination of  $CO_2$  based on the CIE?

(453-455) "We refrain from estimating a CO2 change due to the complex mixing of OM sources. However, the deep-sea benthic  $\delta^{18}$ O record indicates that deep sea temperatures decreased by 1°C in the POIM (Barnet et al., 2019), which is consistent with a modest (20–30%) decline in CO2, assuming a climate sensitivity of 3°C." Given all the work that was done to quantify the various OM sources and degradation, this statement is a bit disappointing (besides, the authors do assign values, e.g., 550-553, where they identify a residual excursion of ~2.5‰, exclusive of degradation processes, or the purported 4‰ CIE measured in phytane, 376). As noted above, even a back-of the-envelope calculation given a few assumptions (or a range of CIE sizes) would be useful to see if a CO2 decline is even a plausible interpretation from the  $\delta^{13}$ C data. Otherwise the entire premise of a CO2 decline is based solely on data separate from this study (deep-sea benthic  $\delta^{18}$ O data and climate sensitivity estimates).

Thanks very much for these comments. To be honest this is an issue that was intensely debated by the co-authors. Some of us were very much in favour in making specific  $CO_2$  determinations, whereas others argued that the uncertainties were too great to provide an estimate. As reviewer 1 notes, we were concerned that any specific estimate would likely be widely cited because data for this interval are so sparse and for this reason, we resiled from including an estimate. However, the reviewer has convinced us that this is a major shortcoming in the paper. Therefore, we have prepared a new section for the revised paper (see Calculating  $CO_2$  in the attached file). Note that the requirement to differentiate between terrestrial and marine OM is simply because the method we employ is based on the  $\delta^{13}C$  of terrestrial OM.

We welcome further comments from the reviewer on this addition to the paper.

**Specific Comments:**

82: "From these analyses, we estimate the magnitudes of the  $\delta^{13}$ C excursion in both primary terrestrial and marine OM and use these values to infer broad changes in the concentration of atmospheric CO2." Where is the calculation of CO2 from the  $\delta^{13}$ C data?

This is now added as noted above.

Many of the geochemical methods are repeatedly simply cited back to Naeher et al. (2019), rather than being reported here. At least, a brief summary of the methods used here would be useful to the reader. For example, some important details on the standards used for IRMS and the analytical precision of these measurements, which may differ from the previous work? This was done for the compound specific work, but would make reading this paper easier as a stand-alone product, without needing to read back to Naeher et al. (2019) for the methods.

Agreed. We will add summary methods to this paper.

The summary paragraph of Section 5.4 "13C enrichment attributable to drawdown of atmospheric  $CO_2$ " lacks any description of how 13C enrichment relates to drawdown of  $CO_2$ .

Yes. Text was transferred to section 6.2 but we realise that it leaves the question hanging, so will amalgamate these two sections.

Conclusions. I think a calculation of  $CO_2$  from the  $\delta^{13}C$  data would go a long way towards bolstering the linkages between  $CO_2$ , cooling, C burial, volcanism, and sea level, etc proposed in the conclusions.

Yes, agreed.

**Technical Corrections:**

Throughout, delta values ( $\delta^{13}C$ ,  $\delta^{18}O$ ) are commonly described as heavy/enriched (or depleted), rather than as being higher/lower. It is my understanding that a sample is enriched (or depleted) in one isotope (e.g.,  ${}^{13}C$ ), but cannot be enriched/depleted in  $\delta^{13}C$  (or  $\delta^{18}O$ ). Some examples of these various permutations are noted here:

17: enriched in  $\delta^{13}C$  --> enriched in  $^{13}C$

19: heaviest  $\delta^{13}$ C values --> greatest  $\delta^{13}$ C values

70:  $\delta^{13}C_{OM}$  value of -20‰, which is ~7‰ heavier --> ~7‰ greater

236/249: more depleted  $\delta^{13}C_{OM}$  values --> lower  $\delta^{13}C$  values

527: depleted  $\delta^{18}$ O values --> lower  $\delta^{18}$ O values

Yes, OK, will correct these and other instances.

**263-264: citation?**

Sure. We will add reference to Rontani and Volkman, 2003 (Rontani, J.-F. and Volkman, J. K.: Phytol degradation products as biogeochemical tracers in aquatic environments, Organic Geochemistry, 34, 1-35, https://doi.org/10.1016/S0146-6380(02)00185-7, 2003.)

308-310: See also Lukens et al. (2019): The effect of diagenesis on carbon isotope values of fossil wood: Geology, v. 47, p. 987–991, https://doi.org/10.1130/G46412.1.

Important reference but only discusses the negative shift in  $\delta^{13}$ C in the first phase of diagenesis.

486: It difficult --> it is difficult

Got it.

\*\*\*\*\*

**Calculating atmospheric CO2**

We have explored the relationship between atmospheric CO2 and C3 plant tissue  $\delta^{13}$ C values (Cui and Schubert, 2016; Cui and Schubert, 2017; Cui and Schubert, 2018; Schubert and Jahren, 2012; Schubert and Jahren, 2018) to estimate atmospheric CO2 concentrations prior to and during Waipawa deposition. The change in  $\delta^{13}$ C ( $\Delta^{13}$ C) per ppm of CO2 follows a hyperbolic relationship (Schubert and Jahren, 2012) and is based on the model of carbon isotope fractionation in plants originally described by Farquhar et al. (1989). This proxy yields an estimate for CO2 that is based on the relative change in  $\Delta^{13}$ C between the time of interest ( $\Delta^{13}C_{(t)}$ ) and the  $\Delta^{13}$ C value at a chosen initial time ( $\Delta^{13}C_{(t=0)}$ ), which is designated as  $\Delta(\Delta^{13}C)$  and expressed as Equation 1:

$$\Delta(\Delta^{13}C) = \frac{[(A)(B)(CO_{2(t)}+C)]}{[A+(B)(CO_{2(t)}+C)]} - \frac{[(A)(B)(CO_{2(t=0)}+C)]}{[A+(B)(CO_{2(t=0)}+C)]}$$
(1)

where A, B and C are curve fitting parameters, and solved for  $CO_2$  at any time t ( $CO_{2(t)}$ ) as Equation 2 (Cui and Schubert, 2016):

$$CO_{2(t)} = \frac{\Delta(\Delta^{13}C) \cdot A^{2} + \Delta(\Delta^{13}C) \cdot A \cdot B \cdot CO_{2(t=0)} + 2 \cdot \Delta(\Delta^{13}C) \cdot A \cdot B \cdot C + \Delta(\Delta^{13}C) \cdot B^{2} \cdot C \cdot CO_{2(t=0)} + \Delta(\Delta^{13}C) \cdot B^{2} \cdot CO_{2(t=0)}}{A^{2} \cdot B - \Delta(\Delta^{13}C) \cdot A \cdot B - \Delta(\Delta^{13}C) \cdot B^{2} \cdot CO_{2(t=0)} - \Delta(\Delta^{13}C) \cdot B^{2} \cdot C}$$

$$(2)$$

The combined uncertainty of parameters used to derived the estimate for atmospheric  $CO_2$  is relatively large and increases with increasing  $CO_2$  (Cui and Schubert, 2016, 2018).

As in Cui and Schubert (2018), we use the latest Paleocene (56.1–56.5 Ma, t=0) as the reference time and adopt the same parameters (Table 1) with some modifications. We exclude an unusually low estimate of 100 ppm for CO2 derived from paleosols by Sinha and Stott (1994) and we base our estimates for the  $\delta^{13}$ C of atmospheric CO2 ( $\delta^{13}$ CCO2) on the method described by Tipple et al. (2010) but recalculated using the smoothed LOESS benthic foraminiferal  $\delta^{13}$ C and  $\delta^{18}$ O curves of Westerhold et al. (2020). [Add equations 3, 4, 5 and 7 of Tipple et al. (2010) here]. For this calculation, we use the temperature equation of Kim and O'Neil (1997) rather than that of Erez and Luz (1983), which is not appropriate for benthic foraminiferal calcite (Hollis et al., 2019). We assume ice-free conditions for this calculation (i.e.,  $\delta^{18}$ Ow = -1‰), while noting that the findings of this study imply the growth of ice sheets during Paleocene episodes. The three time slices used for our  $\delta^{13}$ CCO2 reconstructions are: latest Paleocene (pre-PETM) reference time slice, 56–56.2 Ma; Waipawa organofacies (WOF), 59–59.2 Ma; underlying organofacies (pre-WOF), 59.6–59.8 Ma (Table 1, Figure 1).

We have derived three estimates for the change in CO2 that can be linked to Waipawa deposition. These are based on estimated bulk terrestrial  $\delta^{13}$ C values as well as  $\delta^{13}$ C values for the higher plant biomarkers, odd-numbered HMW *n*-alkanes (C27-C33) and even-numbered HMW fatty acids (C26-C32). For these lipid biomarkers we add 4‰ to the raw  $\delta^{13}$ C values to account for isotope effects during the biosynthesis of *n*-alkyl biomolecules (Diefendorf et al., 2015). Similarly, in the absence of equivalent *n*-alkane and fatty acid data for the latest Paleocene, we subtract 4‰ from the terrestrial reference value, which is derived from a latest PETM coal deposit in northeast China (Chen et al., 2014).

For HMW fatty acids in the mid-Waipara section, the carbon isotope excursion (CIE) from the mean value for underlying facies to the mean value for the main phase of Waipawa deposition is 2.6‰ (Table S6, mean raw values of -31.6 and -29‰). For HMW *n*-alkanes in the Taylor White section, we have argued that the HMW *n*-alkanes in the Waipawa facies have been affected by mixing. If we substitute values from the nearby Angora Road site, we derive a CIE of 3.3‰ based on the average of two OM-rich Waipawa samples from Angora Road (raw value of -27.9‰) and a

single sample from underlying Whangai facies in the Taylor White section (raw value of -30.7; Table S4). Because we cannot be sure of the extent to which the bulk terrestrial  $\delta^{13}$ C values are affected by lignin alteration, we have adopted an intermediate value of 3‰ for the bulk organic CIE. We use the  $\delta^{13}$ C values from the density fractions from the Taylor White section (Section 5.2.1) to derive a value of -21‰ for terrestrial OM in underlying Whangai facies. A CIE of 3‰ implies a value of -18‰ for Waipawa organofacies. As the maximum value for Waipawa organofacies is -16.7‰, this suggests that lignin degradation may only account for ~1‰ of the total excursion.

The three approaches result in significant differences in CO2 estimates, both for Waipawa facies and the underlying facies (Table 1). CO2 estimates range from 208 to 368 ppm for Waipawa organofacies and from 333 to 609 ppm for the underlying facies. This represents a 37–44% decrease in CO2 during Waipawa deposition. This variation in values is to be expected given the many sources of uncertainty related to estimating the magnitudes of the CIEs for each parameter, variability within biomarkers and uncertainties in the calibration itself. Nevertheless, the different approaches yield consistent estimates of a ~40% decrease in CO2 that can be linked to Waipawa deposition. Temperature estimates derived from the benthic foraminiferal compilation indicates global temperature decreased by ~1°C from the pre-WOF to WOF time slices (Figure 1). An accompanying decrease of ~40% in CO2 equates to a decrease of 2.5°C for a halving of CO2). However, as noted above, our temperature calculations assume ice-free conditions. If cooling was associated with ice growth, a portion of the positive shift in  $\delta^{18}$ O should be attributed to this increase in ice volume, which would lead to a smaller decrease in temperature and, therefore, lower climate sensitivity.

Our estimates for  $CO_2$  in the underlying facies are consistent with published estimates for  $CO_2$  in the Paleocene (Figure 1; LOESS curve from Foster et al., 2017, data from sources cited in Foster et al., 2017; Hollis et al., 2019), with best fit shown by terrestrial OM and *n*-alkanes. Whilst we acknowledge our  $CO_2$  estimates rely on several assumptions and some potentially large sources of error, the implication is that  $CO_2$  levels during Waipawa deposition were in the range of 200–300 ppm, i.e., below modern levels and low enough for polar ice sheet growth.

**References**

- Chen, Z., Ding, Z., Tang, Z., Wang, X., and Yang, S.: Early Eocene carbon isotope excursions: Evidence from the terrestrial coal seam in the Fushun Basin, Northeast China, Geophysical Research Letters, 41, 3559-3564, https://doi.org/10.1002/2014GL059808, 2014.
- Cui, Y. and Schubert, B. A.: Quantifying uncertainty of past pCO2 determined from changes in C3 plant carbon isotope fractionation, Geochimica et Cosmochimica Acta, 172, 127-138, https://doi.org/10.1016/j.gca.2015.09.032, 2016.

- Cui, Y. and Schubert, B. A.: Towards determination of the source and magnitude of atmospheric pCO2 change across the early Paleogene hyperthermals, Global and Planetary Change, 170, 120-125, https://doi.org/10.1016/j.gloplacha.2018.08.011, 2018.
- Diefendorf, A. F., Freeman, K. H., Wing, S. L., Currano, E. D., and Mueller, K. E.: Paleogene plants fractionated carbon isotopes similar to modern plants, Earth and Planetary Science Letters, 429, 33-44, https://doi.org/10.1016/j.epsl.2015.07.029, 2015.
- Farquhar, G.D., Ehleringer, J.R, and Hubick, K. T.: Carbon Isotope Discrimination and Photosynthesis, Annual Review of Plant Physiology and Plant Molecular Biology, 40, 503-537, 10.1146/annurev.pp.40.060189.002443, 1989.
- Foster, G. L., Royer, D. L., and Lunt, D. J.: Future climate forcing potentially without precedent in the last 420 million years, Nature Communications, 8, 14845, 10.1038/ncomms14845, 2017.
- Hollis, C. J., Dunkley Jones, T., Anagnostou, E., Bijl, P. K., Cramwinckel, M. J., Cui, Y., Dickens, G. R., Edgar, K. M., Eley, Y., Evans, D., Foster, G. L., Frieling, J., Inglis, G. N., Kennedy, E. M., Kozdon, R., Lauretano, V., Lear, C. H., Littler, K., Lourens, L., Meckler, A. N., Naafs, B. D. A., Pälike, H., Pancost, R. D., Pearson, P. N., Röhl, U., Royer, D. L., Salzmann, U., Schubert, B. A., Seebeck, H., Sluijs, A., Speijer, R. P., Stassen, P., Tierney, J., Tripati, A., Wade, B., Westerhold, T., Witkowski, C., Zachos, J. C., Zhang, Y. G., Huber, M., and Lunt, D. J.: The DeepMIP contribution to PMIP4: methodologies for selection, compilation and analysis of latest Paleocene and early Eocene climate proxy data, incorporating version 0.1 of the DeepMIP database, Geosci. Model Dev., 12, 3149-3206, 10.5194/gmd-12-3149-2019, 2019.
- Schubert, B. A. and Jahren, A. H.: The effect of atmospheric CO2 concentration on carbon isotope fractionation in C3 land plants, Geochimica et Cosmochimica Acta, 96, 29-43, https://doi.org/10.1016/j.gca.2012.08.003, 2012.
- Schubert, B. A. and Jahren, A. H.: Incorporating the effects of photorespiration into terrestrial paleoclimate reconstruction, Earth-Science Reviews, 177, 637-642, https://doi.org/10.1016/j.earscirev.2017.12.008, 2018.
- Sinha, A. and Stott, L. D.: New atmospheric pCO2 estimates from palesols during the late Paleocene/early Eocene global warming interval, Global and Planetary Change, 9, 297-307, https://doi.org/10.1016/0921-8181(94)00010-7, 1994.
- Tipple, B. J., Meyers, S. R., and Pagani, M.: Carbon isotope ratio of Cenozoic CO2: A comparative evaluation of available geochemical proxies, Paleoceanography, 25, 10.1029/2009pa001851, 2010.
- Westerhold, T., Marwan, N., Drury, A. J., Liebrand, D., Agnini, C., Anagnostou, E., Barnet, J. S. K., Bohaty, S. M., De Vleeschouwer, D., Florindo, F., Frederichs, T., Hodell, D. A., Holbourn, A. E., Kroon, D., Lauretano, V., Littler, K., Lourens, L. J., Lyle, M., Pälike, H., Röhl, U., Tian, J., Wilkens, R. H., Wilson, P. A., and Zachos, J. C.: An astronomically dated record of Earth's climate and its predictability over the last 66 million years, Science, 369, 1383-1387, 10.1126/science.aba6853, 2020.

Table 1. Parameters used to calculate atmospheric CO2 and resulting CO2 estimates for Waipawa and underlying organofacies

|                         | Age (Ma)  | A 1 | B 1 | C 2 | $\delta^{13}C\ ^3$ | δ 13 C[CO 2 ] 4 | Δ(Δ 13 C) 1 | $CO_2^2$ | % Decrease 5 | Sensitivity 6 |
|-------------------------|-----------|----------------|----------------|----------------|--------------------|--------------------------------------------------|-----------------------------------|----------|--------------|---------------|
| Terrestrial OM          |           |                |                |                |                    |                                                  |                                   |          |              |               |
| Latest Paleocene (t = 0 | 56.5-56.1 | 28.26          | 0.22           | 23.69          | -22.00             | -5.80                                            |                                   | 385      |              |               |
| OM-rich Waipawa         | 59-58.5   | 28.26          | 0.22           | 23.69          | -18.00             | -5.00                                            | -3.33                             | 208      | 0.37         | 2.7           |
| Underlying facies       | 60-59.5   | 28.26          | 0.22           | 23.69          | -21.00             | -5.50                                            | -0.73                             | 333      |              |               |
| N-alkanes               |           |                |                |                |                    |                                                  |                                   |          |              |               |
| Latest Paleocene (t = 0 | 56.5-56.1 | 28.26          | 0.22           | 23.69          | -26.00             | -5.80                                            |                                   | 385      |              |               |
| OM-rich Waipawa         | 59-58.5   | 28.26          | 0.22           | 23.69          | -23.45             | -5.00                                            | -1.85                             | 270      | 0.44         | 2.3           |
| Underlying facies       | 60-59.5   | 28.26          | 0.22           | 23.69          | -26.72             | -5.50                                            | 1.06                              | 485      |              |               |
| Fatty acids             |           |                |                |                |                    |                                                  |                                   |          |              |               |
| Latest Paleocene (t = 0 | 56.5-56.1 | 28.26          | 0.22           | 23.69          | -26.00             | -5.80                                            |                                   | 385      |              |               |
| OM-rich Waipawa         | 59-58.5   | 28.26          | 0.22           | 23.69          | -25.00             | -5.00                                            | -0.23                             | 368      | 0.40         | 2.5           |
| Underlying facies       | 60-59.5   | 28.26          | 0.22           | 23.69          | -27.60             | -5.50                                            | 1.99                              | 609      |              |               |

Notes

1. From Cui and Schubert (2016).

2. From Cui and Schubert (2018). Latest Palecene CO2 reconstruction based on data sources listed therein, but exluding Sinha and Stott (1994).

3. From Chen et al. (2014).

4. Calculated from Westerhold et al. (2020) using method of Tipple et al. (2010).

5. Percentage decrease in CO2 in Waipawa organofacies

6. Decrease in temperature (°C) with one halving of CO2.

Figure 1. Compilation of early Paleogene variation in deep-sea benthic foraminiferal (a) carbon and (b) oxygen isotopes (LOESS smoothed curves from Westerhold et al. 2020), (c) oxygen isotopebased temperatures, (d) carbon isotope values for atmospheric CO2 and (e) estimates for atmospheric CO2 volume (after Foster et al., 2017; Hollis et al., 2019; LOESS curve from Foster et al., 2017). Horizontal pink lines – hyperthermals; horizontal yellow lines – reference time slices for CO2 determinations.